# Technical note: Measurements and data analysis of sediment-water oxygen flux using a new dual-optode eddy covariance instrument.

Markus Huettel[1], Peter Berg[2], Alireza Merikhi[1]

[1] Department of Earth, Ocean and Atmospheric Science, Florida State University, Tallahassee, FL 32306-4520, USA
[2] Department of Environmental Sciences, University of Virginia, Charlottesville, VA 22904-4123, USA

*Correspondence to*: Markus Huettel (mhuettel@fsu.edu)

**Abstract.** Sediment-water oxygen fluxes are widely used as a proxy for organic carbon production and mineralization at the seafloor. In-situ fluxes can be measured non-invasively with the aquatic eddy covariance technique, but a critical requirement is that the sensors of the instrument are able to correctly capture the high frequency variations in dissolved oxygen
concentration and vertical velocity. Even small changes in sensor characteristics during deployment as caused e.g. by biofouling can result in erroneous flux data. Here we present a dual-optode eddy covariance instrument (2OEC) with two fast oxygen fibre sensors and document how erroneous flux interpretations and data loss can effectively be reduced by this hardware and a new data analysis approach. With deployments over a carbonate sandy sediment in the Florida Keys and comparison with parallel benthic advection-chamber incubations, we demonstrate the improved data quality and data reliability
facilitated by the instrument and associated data processing. Short-term changes in flux that are dubious in measurements with single oxygen sensor instruments can be confirmed or rejected with the 2OEC and in our deployments provided new insights into the temporal dynamics of benthic oxygen flux in permeable carbonate sands. Under steady conditions, representative benthic flux data can be generated with the 2OEC within a couple of hours, making this technique suitable for mapping sediment-water, intra-water column, or atmosphere-water fluxes.

## 1 Introduction

The significant role of sediments in the marine cycles of matter (Walsh, 1988;Johnson et al., 1999;Jahnke, 2010;Bauer et al., 2013) emphasizes the need for reliable benthic flux data. Where currents and benthic photosynthesis influence interfacial flux at the seafloor, the aquatic eddy covariance technique is a preferred tool for determining benthic oxygen fluxes as it permits flux measurements with minimal disturbance of bottom flow and light. This technique derives vertical oxygen flux from time
series of rapid simultaneous measurements of vertical flow velocity changes and associated oxygen changes at a fixed point above the sediment surface. Since its introduction by Berg et al. (2003), the strength of this non-invasive technique has been demonstrated in marine and freshwater settings (Berg et al., 2013;Chipman et al., 2016;Hume et al., 2011;Reimers et al., 2012;Rheuban et al., 2014a;Lorrai et al., 2010;Attard et al., 2019;Rodil et al., 2019) including environments (e.g. permeable sediments, seagrass beds, coral reefs, hard bottoms, sea ice) that pose challenges to other flux-measuring techniques (Berg et
al., 2009;Brand et al., 2008;Crusius et al., 2008;Glud et al., 2010;McGinnis et al., 2014;Berg and Pace, 2017;Long et al., 2013;Long et al., 2012;Berger et al., 2020).

The main challenge of the present aquatic eddy covariance instrumentation is associated with the high frequency oxygen measurements. The microelectrodes typically used for these measurements (sensor tip < 100 μm), break easily during deployments in energetic coastal environments (Chipman et al., 2012;Berg et al., 2017), and their stirring sensitivity and signal drift add further uncertainties to the oxygen data (Gundersen et al., 1998;Holtappels et al., 2015;Reimers et al., 2016). To improve the reliability of the flux measurements, eddy covariance with dual microelectrodes were developed, e.g. (Attard et al., 2014;McGinnis et al., 2011;Rodil et al., 2019;de Froe et al., 2019;Rovelli et al., 2015)

An alternative to the electrochemical microelectrodes are optical sensors (optodes or optrodes) that use the luminescence characteristics of an oxygen-sensitive dye for oxygen concentration measurements (Klimant et al., 1995;Holst et al., 1998;Bittig et al., 2018;Wang and Wolfbeis, 2014). Optodes consume no oxygen and may have very low or no stirring sensitivity (Holtappels et al., 2015;Berg et al., 2017). Compared to microelectrodes, we have observed they are less susceptible to signal drift and keep their calibration over longer time. Because they are not sensitive to sulphides, optodes are the superior sensor in hypoxic environments or near anoxic sediment surfaces. Weaknesses of optodes include the bleaching of the sensor dye over time, their non-linear calibration with decreasing resolution at high oxygen concentrations, and interference of strong light with the optical measurements (Lehner et al., 2015). Although most optodes are more robust than microelectrodes, they can break due to particle collision. In long-term measurements, the bleaching effect can be reduced through interval measurements. The non-linear calibration can be described by a function based on the Stern-Volmer equation (Stern and Volmer, 1919;Wang and Wolfbeis, 2014) that allows reliable conversion of the optode signal to concentration data, but the decreasing resolution at high oxygen concentrations remains. The light interferences typically are only an issue in very shallow water and can be eliminated by protective fibre coatings.

Irrespective of the technology, the readings of the oxygen sensors can be biased by attachment of particles, bacteria or algal cells, which can affect the sensor signal through shielding of the sensor tip and metabolic processes (Smith et al., 2007;Delauney et al., 2010). Mineral particles may be impenetrable to gases, while organic particles may be sufficiently dense or oxygen consuming such that oxygen diffusion through them is reduced, (Zetsche et al.;Ploug and Passow, 2007) thereby decreasing and delaying oxygen transport to the sensing surface. The ensuing increase in the response time of the sensor dampens the oxygen signal and thereby reduces the calculated flux. Berg et al. (2015) explained how the resulting time offset between the oxygen and the velocity data can cause significant over- or underestimation of the flux. The most common particles attaching to sensors may be marine snow particles (Fig. 1a), sticky aggregates of various organic and inorganic particles glued together by extracellular polymeric substances (EPS) (Alldredge and Silver, 1988). Bacteria and phytoplankton cells commonly contained in these particles can cause both, oxygen consumption and oxygen production, thereby affecting the signals of the oxygen sensor and the fluxes calculated from these readings. As an example, we observed oxygen flux increases up to 4.4 mmol $m^{-2}$ $h^{-1}$ caused by photosynthesis and decreases up to -5.2 mmol $m^{-2}$ $h^{-1}$ caused by respiration of microbes contained in marine snow attached to the oxygen sensor.

While substantial biofouling (Fig. 1b) may be obvious through the changes in signal magnitude and dynamics it causes, minor biofouling is not detectable without a reference. Small deviations in the response time and magnitude of the sensor signal, however, can produce large differences between the measured and true flux (Berg et al., 2015). Some biofouling can be reduced by coating the sensor with antibiotics but such treatments cannot prevent the adherence of marine snow or detritus particles (Navarro-Villoslada et al., 2001).

With the advantages of being relatively robust compared to microelectrodes and less expensive, optodes are predisposed to become the preferable sensor-type for aquatic eddy covariance measurements (Chipman et al., 2012;Berg et al., 2016). The recent development of small, programmable oxygen meters with low power consumption that can read fibre optodes with short response times made optode technology suitable and accessible for aquatic eddy covariance measurements. The goals of this study were 1) to develop an eddy covariance instrument with dual optodes that through parallel oxygen measurements allows improved quality control of the oxygen data and thereby more reliable oxygen fluxes, 2) to develop a data evaluation procedure for the dual optode eddy covariance data sets that helps identifying compromised optode signals, 3) to demonstrate the performance of this instrument through deployment in an inner shelf environment with dynamic changes in sediment-water flux.

## 2 Methods

### 2.1 Instrument development

The eddy covariance instrument we developed uses two Pyroscience™ FireSting $O_2$-Mini oxygen meters (specifications listed in Table A1 in Appendix A, now sold in combination with the Pyroscience™ subport (FSO2-SUBPORT)) that read two ultra-high speed Pyroscience™ OXR430-UHS retractable oxygen minisensors (Table A2). The FireSting $O_2$-Mini oxygen meters were supplied with the output power of the ADV (see below). The measuring principle of the Pyroscience™ fibre optodes is based on an indicator dye responding to orange-red light excitation (610-630 nm) and lifetime detection in the near infrared (NIR, 760-790 nm), which reduces cross-sensitivity and interferences (e.g. due to ambient light or fluorescent substances in the water). The ultra-high speed OXR430-UHS optodes achieve response times ($t_{90}$) of 150 to 300 ms (Merikhi et al., 2018) and thus can capture all oxygen fluctuations at the temporal resolution required by the eddy covariance technique (Lorrai et al., 2010;Donis et al., 2015), preventing loss of flux contributions at high turbulence frequencies (McGinnis et al., 2008). The 430 µm diameter optical fibre of these optodes is robust relative to microelectrodes. Our previous field measurements indicated that when operated continuously at a measuring frequency of ~ 8 Hz, the useful lifetime of the OXR430-UHS typically was 3 to 7 days before the signal decreased to a level preventing reliable data interpretation (Huettel, unpubl.). The signal drift over this period was negligible (< 0.03%). The optodes for the in-situ measurements were selected for similar fast response times (< 300 ms) established using the jet-nozzle method introduced by Merikhi et al. (2018). The acoustic Doppler velocimeter

(ADV) used for this eddy instrument was a NORTEK Vector, which is a single-point current meter capable of measuring velocity and current direction in a small measuring volume (14 mm diameter, 14 mm height (user-specified)), at rates up to 64 Hz (Table A3). Together with the current flow measurements, the Vector records pressure and temperature, as well as the compass direction and tilt of the instrument. The internal data logger of the Vector stored the current velocity data simultaneously with the two analogue signals produced by the O2Mini oxygen meters. An external battery with a capacity of 200 Wh connected to the Vector provided power for continuous measurements of up to one week duration.

The instruments were mounted on a tripod (width 120 cm, height 100 cm, Fig. 2), made of rectangular 304-stainless steel tubes (2 cm x 2 cm cross-section), with legs consisting of stainless steel rods, 1.3 cm in diameter with 20 cm diameter base plates. An extension arm held the ADV in the centre of the frame. The underwater housing (AGO Environmental Electronics) containing the oxygen meters with supply voltage regulator (Dimension Engineering) and the external battery pack (4 x NORTEK Lithium-Ion 12 V, 50 Wh) were attached to the horizontal upper bar of the tripod. All electrical cables used Impulse™ wet pluggable micro inline connectors. The two optodes were linked through two custom-made (Huettel) underwater housing fibre-feed-through plugs with standard ST-connectors to the FireSting O2Minis. A stainless steel rod (8 mm diam.) with adjustable holders and aligned with the X-direction of the ADV, positioned the two optodes parallel to each other and at a 45-degree downward angle. The sensing tips of the optodes were 10 mm apart from each other and located at 30 mm horizontal distance from the lower edge of the ADV measuring volume (Fig. 2b). This distance prevents any disturbance of the flow in the measuring volume of the ADV and any interference with the acoustic pulses of the Vector. A PAR-light sensor (Odyssey® Submersible Photosynthetic Active Radiation Logger) installed above the ADV logged light intensity at 5 minute intervals throughout the deployments. An Aanderaa Seaguard RCM multisensor probe, installed with its sensors at the same height as the ADV measuring volume at 5 m distance from the tripod, recorded oxygen and temperature reference data.

**2.2 Field tests**

The performance of the 2OEC was tested through three deployments on 14-15 August 2013 (Case C), 16-17 August 2013 (Case A), and 10-11 April, 2014 (Case B) in a subtropical inner shelf environment with relatively constant salinity (35-36) and temperatures (April: $25° \pm 0.8°C$, August: $30° \pm 0.5°C$) approximately 9 km south of Long Key in the Florida Keys (24° 43.52'N, 80° 49.85'W). The site was located at $9 \pm 1$ m water depth near the centre of a large flat carbonate platform covered with coral sand. The unobstructed, fairly steady current flows across the platform and the relatively uniform surface roughness (ripple topography < 10 cm) produced similar turbulent diffusivity throughout the deployments. The highly permeable carbonate sand (permeability: $k = 3 \times 10^{-11} \pm 0. 2 \times 10^{-11}$ m$^2$) had a median grain size of 440 µm and was inhabited by microphytobenthos (2-6 µg Chl. a g$^{-1}$ sed. dw) and sparsely distributed (< 20 m$^{-2}$) *Halimeda* sp. Macroalgae (Fig. 2A). In the clear water (Turbidity < 8 NTU) light intensities at the seafloor reached up to 300 µE m$^{-2}$ s$^{-1}$. The current flow conditions during all deployments were moderate (average mean flow velocity 5 to 20 cm s$^{-1}$, significant wave height < 0.7 m), and the weather was generally sunny with some scattered clouds. Prior to the deployments, the oxygen optodes were calibrated in

ambient seawater (water bubbled with air or with sodium sulphite addition), with the calibration data stored on the Vector logger. The measuring volume of the ADV was adjusted to be ~35 cm above the sediment-water interface. SCUBA divers positioned the instrument at the seafloor such that the Vector's X-direction was aligned with the main bottom flow direction, which was in northeast-southwest direction. The instrument was typically deployed in the morning at 9:00-10:00 and retrieved

24 h later. During the first hour after deployment, no flux data were collected to allow temperature adjustment of the instruments. Before downloading the data from the Vector, the calibration of the oxygen sensors was repeated and stored with the data file.

## 2.3 Data processing

Velocity data with acoustic beam correlations < 50% were replaced through linear interpolation of the neighbouring velocity

values. Oxygen data were not cleaned or despiked prior to flux calculations. Oxygen fluxes were calculated using EddyFlux 3.2 software package (P. Berg) as follows: Vertical velocity data and oxygen concentration data were reduced from 64 Hz to 8 Hz through averaging, which lessened data noise while maintaining sufficient resolution for resolving high-frequency eddies. The fluctuating component of the oxygen concentrations was determined through Reynolds decomposition, i.e. oxygen base concentrations were determined for 15 min intervals through linear detrending and subtracted from the instantaneous oxygen

data to arrive at the instantaneous oxygen fluctuations $O_2'$. Instantaneous vertical velocity changes $V_z'$ were determined through Reynolds decomposition analogous to the oxygen fluctuations. The time lag caused by the 30 mm horizontal distance between flow and oxygen measurement locations were corrected according to Berg et al. (2015) through applying time shift corrections that yielded most negative (night) or most positive (day) cross-correlations of the oxygen fluctuation and vertical movement. Oxygen fluxes then were calculated by averaging over time the product of instantaneous oxygen fluctuation and

instantaneous vertical velocity change: $O_2 Flux = \overline{O_2' \times V_z'}$ (Berg et al., 2003). At our measuring height of 35 cm above the seafloor, the diurnal fluctuation in mean water column oxygen concentration can result in substantial changes in the oxygen inventory of the water column below the measuring volume, which can bias the local eddy flux measurements. To correct for this effect, an oxygen storage term, calculated as $\int_0^h dC/dt\ h$, was subtracted from the measured eddy flux to determine the benthic oxygen flux (dC/dt = change of the average oxygen concentration over time, calculated through linear detrending of

the measured oxygen data over 15 minute intervals, h = height of the measuring volume)(Rheuban et al., 2014b). Acceleration or deceleration of current flows can alter the oxygen concentration profile and thereby modulate vertical flux (Holtappels et al., 2013). Our measurements indicated that the temporal flux variations caused by transient velocity changes largely cancelled out over time (Rheuban et al., 2014b), and a correction for transient velocity changes was not applied. For the comparison of the temporal evolution of the fluxes that were determined using the recordings of the two optodes, we calculated the cumulative

fluxes over the duration of the deployments. The slopes of the increasing cumulative fluxes during daylight and decreasing cumulative fluxes during nighttime were assessed for hourly time intervals. Standard deviations of the fluxes reflect the deviations between three hourly slope determinations. All error estimates are reported as ±1 standard deviation.

## 2.4 Advection chamber deployments

In August 2013, 3 advection chambers were deployed parallel to the eddy covariance instrument to allow comparison with an independent flux data set produced by a different method. Benthic advection chambers present an in-situ incubation technique that can account for some of the current and light effects influencing benthic flux (Janssen et al., 2005a;Huettel and Gust, 1992b). The rotation of a stirring disk (15 cm diam.) within these cylindrical chambers (30 cm height, 19 cm inner diameter) produces a radial pressure gradient at the surface of the enclosed sediment that is similar in magnitude to the pressure gradient generated by bottom currents interacting with present ripple topography (Huettel and Rusch, 2000). For the deployments at our study site, the pressure gradient in the chambers was set to 0.2 Pa cm$^{-1}$, corresponding to the gradient produced by a 10 cm s$^{-1}$ bottom flow deflected by a ripple of 7 cm height (Huettel and Gust, 1992a). In highly permeable sediments, the pressure gradient in the chamber causes pore water flow through the surface layer of the enclosed sediment, thereby mimicking the pore water exchange occurring in the surrounding rippled seabed. The transparent chamber and stirring disk allow penetration of light to the enclosed sediment (~10% loss in PAR through light attenuation caused by the acrylic), permitting benthic photosynthesis in the chamber. The acrylic cylinder of the chambers was pushed 12 cm into the sand sediment, resulting in a chamber water volume of 5 L. A Hach Rigid O2 Optode mounted in the chamber lid collected oxygen concentrations at 15 minute time intervals. The fluxes were calculated from the changes of the oxygen concentration in the water column of the chamber over time. Chamber incubations ran for 24 h, then the lid was opened to allow re-equilibration with the ambient water before starting the next measurement cycle. Although flow, light and water composition changes within the chamber are not identical to the external conditions and cause an inherent bias, the daily fluxes measured by these chambers are considered to be close (within a factor ~2) to the true fluxes (DeBeer et al., 2005;Cook et al., 2007;Janssen et al., 2005a), and this technique has been deployed successfully in numerous investigations of shallow permeable sediments (Huettel and Gust, 1992b;Eyre et al., 2013;Eyre et al., 2018;Glud, 2008;Cyronak et al., 2013;Santos et al., 2011;Janssen et al., 2005b).

## 3 Results

### 3.1 Instrument deployments

The 2OEC improves the reliability of measured fluxes. The deployment of 16-17 August 2013 (Case A, Fig. 3) was characterized by moderate bottom currents averaging $3.6 \pm 2.2$ cm s$^{-1}$ (35 cm above sediment) with sustained peak velocities of 8.0 cm s$^{-1}$ (Fig. 3A) and relatively low light intensities at the seafloor < 100 µE m$^{-2}$ s$^{-1}$ during daytime hours (Fig. 3E). The good agreement of the independent O$_2$ readings of both fibre optodes and the Seaguard reference optode (Fig. 3B) implied that the optodes maintained their calibration throughout the deployment. Identical corrections were applied to P and Q optode data sets when calculating fluxes (Fig. 3C), which included corrections for change in average water oxygen concentration, time lag, and wave rotation. The conformity of the 15 min cumulative fluxes calculated from the two fibre optode signals (Fig. 3D) and the agreement of the cumulative flux curves over the time course of the deployment (Fig. 3E) corroborated the flux estimates.

The flux increase during daylight and decrease during nighttime could be represented well by linear regression ($R^2 > 0.9$). The

slopes of the cumulative flux curves over the time course of the deployment (Fig. 3E) revealed daytime fluxes of $3.4 \pm 0.6$ mmol m$^{-2}$ h$^{-1}$ (P) and $3.4 \pm 0.4$ mmol m$^{-2}$ h$^{-1}$ (Q) and nighttime fluxes of $-1.3 \pm 0.9$ mmol m$^{-2}$ h$^{-1}$ (P) and $-1.6 \pm 0.9$ mmol m$^{-2}$ h$^{-1}$ (Q). The close agreement between the fluxes calculated from the two optode signals supported an average daytime flux of $3.4 \pm 0.7$ mmol m$^{-2}$ h$^{-1}$ (P, Q) and nighttime flux of $-1.4 \pm 1.3$ mmol m$^{-2}$ h$^{-1}$ (P, Q) for Case A.

Analysis of the differences between optode P and optode Q based fluxes indicated that changes in environmental settings as well as changes in optode characteristics produced the discrepancies. Larger differences between the P and Q 15 minute fluxes were observed during daytime and when fluxes were near zero and changing direction at sunset (Fig. 4A). Patchy distribution of microphytobenthos and its photosynthetic oxygen production may result in a more uneven oxygen distribution in the bottom currents during daytime (Bartoli et al., 2003;Jesus et al., 2005;MacIntyre et al., 1996). Likewise, the patchy distribution of

macrofauna and its activity peak near sunset (Wenzhofer and Glud, 2004) may be responsible for enhanced heterogeneity in the oxygen distribution in the bottom currents and ensuing larger differences between the parallel-measured fluxes at sunset. Figure 4B-F depict the effects of the corrections that were equally applied to optode P and Q data sets to account for changes in environmental parameters and time lag error when calculating the respective fluxes. Corrections for instrument tilt were not required for our three deployments, and rotating the average flow velocity vectors did not produce significant changes in the

fluxes. During the first 4 hours of the deployment, the raw, unprocessed cumulative fluxes (no corrections applied) derived from both optode signals were nearly identical before differences increased (Fig. 4B). Correction for temporal change in the average water oxygen concentration (Fig. 4C), led to slight rate increases in the cumulative fluxes during the day as well as during the night. The correction for time lag between current flow and oxygen signal had an effect mostly during the last 7 h of the deployment (Fig. 4D), possibly due to a minor growth of biofilm on the optodes. Correction for wave rotation caused a

small rate increase in fluxes, which was more pronounced during nighttime (Fig. 4E). Simultaneous application of the above corrections resulted in a nearly perfect agreement between the cumulative fluxes calculated from the two optode signals (Fig. 4F). Temporal flux variations caused by transient velocity changes did not have a significant impact on the cumulative flux as indicated by the good agreement between the fluxes derived from the eddy covariance measurements and those recorded in parallel benthic advection chamber measurements as reported below.


The parallel optode measurements confirmed short-term changes, e.g. the concentration step in the oxygen record at 18:14, caused by the change of the tide (high tide: 18:16) and associated change in flow direction. The slower Seaguard oxygen sensor did not pick up this abrupt step. The temporarily increased benthic oxygen consumption near 20:00, coinciding with sunset, may have resulted from decomposition of highly degradable photosynthesis products accumulating in the sediment during

daytime (Koopmans et al., 2020) and the aforementioned activity burst of the macrofauna at sunset (Wenzhofer and Glud, 2004). The parallel measurements also confirmed transient flux changes, e.g. a ~30 minute period of reduced light after 14:00 temporarily lowered fluxes by ~45 mmol m$^{-2}$ d$^{-1}$. The ~90 minute $O_2$ concentration and flux dip at 2:45 – 4:15 was caused by

a reversing shift in the main flow direction (48° ➔ 71° ➔ 47°), which changed the origin of the water reaching the sensors and thereby the footprint area interrogated by the instrument (Berg et al., 2007). A correlation between 15 min $O_2$ fluxes and

average water $O_2$ concentration changes (up to -0.16 mmol m$^{-2}$ h$^{-1}$/mmol l$^{-1}$ h$^{-1}$) supported that transient $O_2$ changes had to be corrected for when calculating the benthic flux. The weak correlation between 15 min $O_2$ fluxes and short-term mean current flow changes (-0.03 mmol m$^{-2}$ h$^{-1}$/cm s$^{-1}$ h$^{-1}$) indicated that transient current changes had a relatively small effect on the calculated flux.

The 2OEC allows detection of compromised flux data. During the deployment in April 2014 (Case B, Fig. 5), bottom currents were higher compared to Case A, averaging $8.4 \pm 2.8$ cm s$^{-1}$ with sustained peak velocities reaching 16 cm s$^{-1}$ (Fig. 5A). In this deployment, the Seaguard instrument was installed at a greater distance (~10 m) from the eddy covariance instrument, resulting in larger discrepancies between the signals of the fibre optodes and the planar optode of the Seaguard instrument, nevertheless the Seaguard data confirmed the magnitude and main trends of the fibre optode $O_2$ concentrations (Fig. 5B). Four

hours into the deployment, the signals of optode P started to deviate from those of optode Q, culminating in maximum differences in the respective 15-min fluxes at 17:00-17:15 (Fig. 5C, D). A comparison of the cumulative cospectra for that period (Fig. 6) indicated that the P optode may have been compromised through attachment of a marine snow particle containing $O_2$-producing organisms. The steeper increase of the P optode 17:00 curve (Fig. 6) at the dominant wave frequency (0.2-0.3 Hz), suggested that wave orbital motion enhanced the flux, possibly by producing oscillating movement of the particle

attached to the sensor tip. Past 18:00, the cumulative fluxes based on P and Q signals agreed again (Fig. 5E), suggesting that the particle was washed off the sensor. After excluding the compromised data collected between 14:00 and 18:00, the fluxes calculated based on the two optode signals agreed well (daytime: $3.4 \pm 0.6$ mmol m$^{-2}$ h$^{-1}$ (P), $3.3 \pm 0.3$ mmol m$^{-2}$ h$^{-1}$ (Q), nighttime $-0.9 \pm 0.1$ mmol m$^{-2}$ h$^{-1}$ (P), $-0.9 \pm 0.7$ mmol m$^{-2}$ h$^{-1}$ (Q); daytime average $3.3 \pm 0.7$ mmol m$^{-2}$ h$^{-1}$ (P, Q), nighttime average $-0.9 \pm 0.7$ mmol m$^{-2}$ h$^{-1}$ (P, Q)).


The 2OEC can reduce data loss as exemplified by the Case C (Fig. 7, 14-15 August 2013 deployment). Although the oxygen concentrations recorded by the 2OEC and the Seaguard optodes agreed during the deployment (Fig. 7B), optode P was compromised over extended time periods likely due to a particle caught by the sensor. After almost identical fluxes during the first hour (P: $4.0 \pm 0.4$ mmol m$^{-2}$ h$^{-1}$, Q: $4.1 \pm 0.3$ mmol m$^{-2}$ h$^{-1}$, Fig. 7E), the cumulative optode P flux decreased relative to

the optode Q based flux, despite ongoing benthic photosynthetic oxygen production. This decline in P cumulative flux levelled out at 20:00 and remained steady until 22:00 before the trajectories of the two cumulative fluxes matched again. The following good agreement between P and Q cumulative fluxes between 22:00-5:00 (P: $-3.5 \pm 0.1$ mmol m$^{-2}$ h$^{-1}$, Q: $-3.6 \pm 0.3$ mmol m$^{-2}$ h$^{-1}$, Fig. 7E) indicated that sensor P resumed normal operation. Such a sensor recovery can be observed when water currents remove particles that had attached to the sensor disturbing its signal. The identification of the drop in cumulative P flux as an

artefact was supported by the comparison with sensor Q, which produced the typical circadian cumulative flux pattern with a steady increase during the light phase until sunset and decrease thereafter throughout the dark phase. After 5:00, still during

dark conditions, the increase in cumulative P flux and divergence from the cumulative Q flux suggested that sensor P then lost its calibration, which occurs when the sensor loses some of the dye coating that produces the signal (e.g. through particle impact). The temporary good agreement of the cumulative fluxes based on P and Q optode readings permitted salvaging sections of the flux record and thereby allowed at least rough estimates for day and nighttime fluxes for this deployment. (daytime: $4.3 \pm 2.6$ mmol m$^{-2}$ h$^{-1}$, nighttime: -3.2 $\pm$ 0.6 mmol m$^{-2}$ h$^{-1}$, Fig. 7D). The parallel chamber deployments supported these estimates.

## 3.2 Differences between P and Q fluxes and comparison with advection chamber fluxes

After exclusion of flux intervals compromised by biofouling (Case A: no exclusion, Case B: 14:00-18:00, Case C: 12:00-22:00, 5:00-9:00), the differences between P and Q optode fluxes derived from the slopes of the cumulative flux curves (Figs. 3E, 5E, 7E) averaged 2.3%, -0.1% and -4.7% during daytime and 1.7%, 16.2% and -3.2% during nighttime, for the three deployments respectively. These smaller than 20% differences between P and Q optode fluxes strengthened the flux estimates. Fluxes determined with the 2OEC further were supported by the fluxes measured with the advection chambers conducted parallel to the eddy covariance measurement during the August 2013 deployments (Cases A and C, Fig. 8). The average chamber daytime fluxes for the two deployments ($3.9 \pm 3.0$ mmol m$^{-2}$ h$^{-1}$) were similar to the respective eddy covariance fluxes ($3.7 \pm 0.9$ mmol m$^{-2}$ h$^{-1}$) (Fig 8C), although the chamber nighttime fluxes (-3.4 $\pm$ 0.8 mmol m$^{-2}$ h$^{-1}$) exceeded those of the eddy covariance instrument (-2.5 $\pm$ 1.3 mmol m$^{-2}$ h$^{-1}$) by factor 1.4 (Fig 8C). This discrepancy was caused by the smaller nighttime fluxes recorded by the 2OEC during the second August 2013 deployment (Case A, Fig 8B), however, the differences between average eddy and average chamber fluxes were statistically not significant (Fig 8C).

## 4 Discussion

The small and rapid changes in concentration and flow the aquatic eddy covariance instrumentation must record for accurate flux determination make the technique sensitive to even small disturbances affecting the measuring process (Reimers et al., 2012). By using two oxygen sensors recording in parallel, the 2OEC allows detection of measuring artefacts and thereby can enhance the reliability of the flux determinations. The functionality of the 2OEC and the ranges of fluxes it recorded were supported by the general agreement between the 2OEC fluxes and advection chamber fluxes measured parallel to the eddy flux recordings.

A perfect conformity of eddy and chamber fluxes cannot be expected due to fundamental differences between the open, non-invasive eddy covariance and the closed, invasive chamber measuring principles. Although marine sandy sediments as those investigated here may appear homogeneous, bottom current patterns and patchy colonization by e.g. algae and macrofauna cause some spatial and temporal variability in organic matter content and associated microbial metabolic activities that may influence interfacial solute fluxes (Kourelea et al., 2004;Ricart et al., 2015;Wilde and Plante, 2002;Attard et al., 2019). The

fluxes recorded by the 2OEC originated from a sediment surface area of approximately 40 m$^2$ upstream the instrument (Berg et al., 2007), and the location of this footprint area moved with flow direction. The 2OEC therefore integrated some of the natural spatial variability of the flux, and the movement of the footprint area as well as changes in bottom flow velocity are reflected in the measured fluxes. In contrast, each chamber enclosed the same surface area of about 0.03 m$^2$ and applied the same constant pressure gradient at the sediment surface throughout each deployment. The exclusion of flow variations, temporal water composition changes, and some of the light affected the fluxes measured by the chambers. These differences between the two flux-measuring techniques may explain some of the discrepancies in dark fluxes observed between 2OEC and chambers during the 16-17 Aug 2013 deployment (Fig. 8 B) and emphasize the need for including natural bottom currents and light, and integrating over larger surface areas when assessing benthic interfacial fluxes.

The 2OEC improves detection of sensor fouling. This is significant as the most common and most unnoticed cause for aquatic eddy covariance measuring errors likely is the attachment of marine snow particles or biofilms to the solute sensor (extreme case shown in Fig. 1b). Through physical separation of the sensing surface from the water, such fouling increases sensor response time, which decreases the measured rates of oxygen change and the temporal alignment of oxygen and flow data. Furthermore, biological and chemical reactions in such organic coatings can produce or consume oxygen and thereby compromise flux calculations. As the growth of a biofilm on the sensor may be gradual, the detection of the onset of flux bias caused by biofouling may be impossible in a single-sensor instrument. A very good agreement of the cumulative fluxes calculated from the two 2OEC optodes as seen in Case A is a strong indication that the sensors measured correctly (Fig. 3), while differences between the cumulative fluxes as observed in Case C are indicative of sensor malfunction (Fig. 7E).

The comparison of the cumulative fluxes can reveal even short or small deviations of the sensor signal as e.g. caused by a temporary attachment of a marine snow particle (Fig. 5E). If one of the two parallel measuring sensors showed a temporary increase or drop in oxygen as found in Cases B and C, we attributed this to the biofouling of that sensor, and in-situ inspections of the sensors confirmed biofouling. Marine snow was present in the water at the study site partly due to its proximity to coral reefs that release mucus to the water (Wild et al., 2004). In Case B, unusual contributions to optode P fluxes in the wave frequency band (0.2 to 0.3 Hz) that were not mirrored in optode Q, identified optode P as compromised starting at 15:00 for a ~3 h duration (Fig. 6). A marine snow particle with photosynthesizing organisms attached to the tip of the oxygen sensor P may have caused the erroneous flux estimates. Oxygen concentration in the centre of such aggregates during light conditions can be increased by 85 % relative to the surrounding water (Ploug and Jorgensen, 1999), or even by 180% within the sticky millimetre-size gelatinous colonies of *Phaeocystis spp.*, a common global bloom-forming phytoplankton organism (Ploug et al., 1999). The movement of such an attached photosynthesizing particle by wave orbital motion can synchronize vertical current flow oscillations and the effect of the particle on the oxygen reading (e.g. increased oxygen due to photosynthesis) and thereby lead to erroneous flux estimates (Fig. 9).

In single sensor eddy covariance instruments, obvious temporary sensor malfunctions typically flag long sections or the entire deployment as compromised because it is difficult to determine with certainty when and for how long the sensor reading has been biased. The relatively frequent occurrence of sensor fouling therefore causes substantial losses in data, time and costs. The dual sensor approach can reduce such losses because it allows identifying periods of unbiased measurements within partly compromised data records.

The reliability of the flux data hinges on unbiased sensor data that can capture temporal variability of current flow and the oxygen it carries, which may change as rapidly as 1-3 Hz (McGinnis et al., 2008;Kuwae et al., 2006). The ADV used in the 2OEC can produce calibrated current data non-invasively at a frequency of 64 Hz, while the fibre optode has a slower response time (200-300 ms, (Merikhi et al., 2018)), and its placement near the ADV measuring volume could affect current flow measurements and thereby bias the flux calculations. A cylindrical sensor placed in the path of the flow upstream the ADV measuring volume can shed a vortex street thereby compromising the flow in the measuring volume and the flux estimates based on the flow measurements. Depending on the flow Reynolds number, such vortices may extend between 5 to 20 times the diameter of the cylinder downstream the sensor (Green, 2012). By using the Pyroscience fibre optode for the 2OEC, one of the smallest and fastest oxygen sensors presently available, potential errors caused by the disturbance of the flow and interference with the acoustic pulses of the Doppler velocimeter can be avoided. At the turbulent Reynolds numbers typical for our study site ($4000 < Re < 110000$), the vortices shed by the 430 μm fibre exposed to the water currents extend between 2 to 10 mm downstream of the fibre. Since the tips were placed at 30 mm horizontal distance from the lower edge of the ADV measuring volume, turbulence caused by fibre-flow interaction could not reach the ADV measuring volume. Similarly, the sensor tips at that distance did not interfere with the acoustic pulses of the ADV, and when initially positioning the optode tips, we confirmed that the fibres did not cause any disturbances in the ADV signal.

The Pyroscience fibre optode used with the 2OEC is one of the smallest and fastest oxygen sensors available, and a comparison with the most common oxygen sensors presently utilized for aquatic eddy covariance (Table A4) favours the selection of this sensor for many field settings. For this comparison, three eddy covariance instruments equipped with either (1) one Unisense electrochemical microelectrode (Berg et al., 2019), (2) one JFE Advantech Rinko planar optode (Berg et al., 2016), or (3) one Pyroscience fibre optode were deployed side by side (i.e. 10 m spacing) at our study site 3-4 December 2016. All instruments used the same type of tripod and ADV and the oxygen sensors were mounted at a 45-degree downward angle as described for the 2OEC. The three different sensors measured very similar fluxes when the current flow approached the sensor tips from the front as shown in Fig. 10a, burst 11 to 28. This changed when the flow approached the sensors from the back. The RINKO sensor under such flow conditions may self-shade its planar optode, which may result in an underestimation of the fluxes at higher frequencies (0.1-1.0 Hz, Fig. 10b) as seen in burst 1 to 9 (Fig. 10a), and possibly also disturb the flow in the Vector flow measuring volume. In environments with unidirectional current, however, the Rinko sensor facing the flow can produce very clean flux data due to its relatively large sensing surface (Berg and Pace, 2017). There were no significant differences

between the fluxes based on the fibre optode and the microelectrode for the reversed flow, supporting the choice of the sturdier fibre optodes for oxygen measurements with aquatic eddy covariance instruments in settings with changing flow direction.

**5 Conclusions**

We propose using the agreement/disagreement between the fluxes calculated from the signals of two independently measuring
optodes as a tool to assess the quality of the measured fluxes. The nearly identical cumulative fluxes calculated from the two optodes in our August (Case A) and April (Case B) deployments strongly imply that the dynamics of the fluxes were measured accurately by the system. Likewise, the near linearity of the cumulative flux increase during daytime and decrease during nighttime (Figs. 3e, 5e) and the very similar slopes of these cumulative flux curves support that the measurements recorded representative fluxes. The good agreement of the fluxes measured with the eddy covariance instrument and the fluxes measured
independently with a very different method (advection chambers, Fig. 8a, b, d) indicate that the magnitudes of the fluxes recorded by the 2OEC were correct. The deployments of the 2OEC in the Florida Keys sandflat revealed that biofouling frequently affects the aquatic eddy covariance measurements even in such an oligotrophic environment with very clear water containing low amounts of phytoplankton, bacteria and particles. Further developments of the aquatic eddy covariance technique therefore may benefit from installations of devices that monitor (e.g. with a camera) and reduce or prevent biofouling
(e.g. through a cleaning mechanism). This project intended improving the reliability of the aquatic eddy covariance technique and the procedures of data analysis in order to promote this powerful technique. The advantages of 2OEC flux measurements over invasive measurements (e.g. benthic chambers) may be most significant for deployments in continental margins. The magnitudes of biogeochemical benthic processes increase with decreasing water depth, with benthic fluxes reaching highest rates and dynamics in the shelf environment (Huettel et al., 2014;Middelburg and Soetaert, 2004;Jahnke, 2010;Bauer et al.,
2013;Reimers et al., 2004). Here light, bottom currents and waves may strongly influence benthic fluxes (Gattuso et al., 2006). The relatively high fluxes and daytime oxygen release recorded at our oligotrophic sandy study site, supported by flux measurements from similar subtropical and tropical carbonate environments (Bednarz et al., 2015;Rao et al., 2012;Wild et al., 2009;Wild et al., 2005;Glud et al., 2008), emphasize the need for instrumentation that reliably can take light and flow at the seafloor into account when measuring benthic fluxes. The 2OEC is a powerful tool that meets these requirements, and its
relatively high temporal resolution can provide new insights into the dynamics of benthic oxygen flux.

## 6 Appendix A

**Table A1: Specifications of the Pyroscience™ FireStingO$_2$-Mini oxygen meter. This oxygen meter recently has been replaced by the Pyroscience™ PICO-O2, which is similar to the FireStingO2-Mini but more compact. The FireStingO2-Mini oxygen meter is still available in combination with the Pyroscience™ subport (FSO2-SUBPORT).**

| Pyroscience™ FireStingO$_2$-Mini | Single sensor module, |
|---|---|
| Oxygen port | 1 fibre-optic ST-connector |
| Temperature port | 4-wire PT100, -30°C-150°C, 0.02°C resolution, ±0.5°C accuracy |
| Dimensions and Weight | 67 x 25 x 25 mm, 70 g |
| Measuring principle | Luminescence lifetime detection (REDFLASH) |
| Excitation Wavelength | 620 nm (orange-red) |
| Emission wavelength | 760 nm (NIR) |
| Maximum sampling rate | 20 Hz |
| Interface | Serial interface (UART), ASCII communication protocol |
| Analog output | 0 - 2.5 V DC, 14 bit resolution (the meter presently uses half the potential voltage range of the Vector analogue input, but developments are in place to increase this to the full range (0-5V). |
| Power requirements | Max. 70 mA at 5 V DC from USB (typ. 50 mA) |

**Table A2: Specifications of the Pyroscience™ OXR430-UHS retractable oxygen minisensors**

| Optical O$_2$ fibre sensor type | Pyroscience™ OXR430-UHS |
|---|---|
| Fibre diameter | 430 μm |
| Optimal measuring range | 0-720 μmol l$^{-1}$ |
| Maximum measuring range | 0 - 1440 μmol l$^{-1}$ |
| Response time | < 0.3 s |
| Detection limit | 0.3 μmol l$^{-1}$ |
| Resolution at 1% O$_2$ | 0.16 μmol l$^{-1}$ |
| Resolution at 20% O$_2$ | 0.78 μmol l$^{-1}$ |
| Accuracy at 1% O$_2$ | ± 0.31 μmol l$^{-1}$ |
| Accuracy at 20% O$_2$ | ± 3.13 μmol l$^{-1}$ |
| Temperature range | 0 - 50°C |

**Table A3: Specifications of the NORTEK Vector acoustic Doppler velocimeter**

| Sensor | Range | Accuracy | Precision/Resolution |
|---|---|---|---|
| Velocity | ±0.01, 0.1, 0.3, 1, 2, 4, 7 m s$^{-1}$ | ± 0.5% | ± 1% |
| Pressure | 0-20 m (shallow water version) | 0.5% (full scale) | < 0.005% of full scale |
| Temperature | -4 to +40 °C | 0.1 °C | 0.01 °C |
| Compass | 360º | 2º | 0.1º |
| Tilt | < 30° | 0.2° | 0.1° |

**Table A4. The specifications of the oxygen microelectrode, Rinko planar optode and Pyroscience fibre optode**

| Sensor | OX-10 fast (μm) | RINKO EC | OXR430-UHS |
|---|---|---|---|
| Type | Microelectrode | Planar optode | Fibre optode |
| Manufacturer | Unisense | JFE-Advantech | Pyroscience |
| Measurement principle | Electrolytical reduction | Phosphorescence | Phosphorescence |
| Tip diameter (μm) | 10 | 12000 | 430 |
| Response time (90%) (s) | < 0.3 | < 0.5 | < 0.3 |
| Range (% air saturation) | 0-200 | 0-200 | 0-500 |

## 7 Data availability

The current flow and oxygen data collected with the 2OEC during the August 2013 and April 2014 deployments are available at the Biological and Chemical Oceanography Data Management Office (BCO-DMO, https://www.bco-dmo.org/) under DOI: 10.26008/1912/bco-dmo.812523.1. Suggested Citation: Huettel, M., Berg, P., Merkihi, A. (2020) Current flow and oxygen concentrations recorded by the 2OEC-instrument in the Florida Keys from August 2013 and April 2014. Biological and Chemical Oceanography Data Management Office (BCO-DMO). (Version 1) Version Date 2020-05-2. DOI:10.26008/1912/bco-dmo.812523.1 [access date]

## 8 Author contributions

MH designed and assembled the 2OEC instrument, MH, PB and AM deployed the 2OEC, and analysed the data. MH prepared the manuscript with contributions from all co-authors.

## 9 Competing interest statement

The authors declare that they have no conflict of interest.

## 10 Acknowledgments

We thank the staff of the FIO Florida Keys Marine Laboratory for help with instrument deployments and sample collection. We also would also like to thank DSO Chris Peters and the staff of the FSU coastal marine lab for providing SCUBA support. Brian W. Wells, Pascal Brignole, Lee Russell, and Natalie Geyer helped with the chamber deployments. We would like to extend our special thanks to referees Clare Reimers and Karl Attard. Their questions and comments greatly helped to improve the first version of this manuscript. The research was conducted under NOAA permit FKNMS-2012-137-A2 and was supported by NSF grants OCE-1334117, OCE-1851290, and OCE-1061364.

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

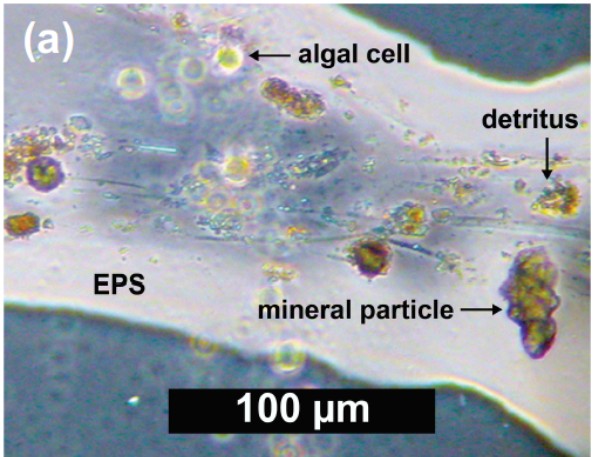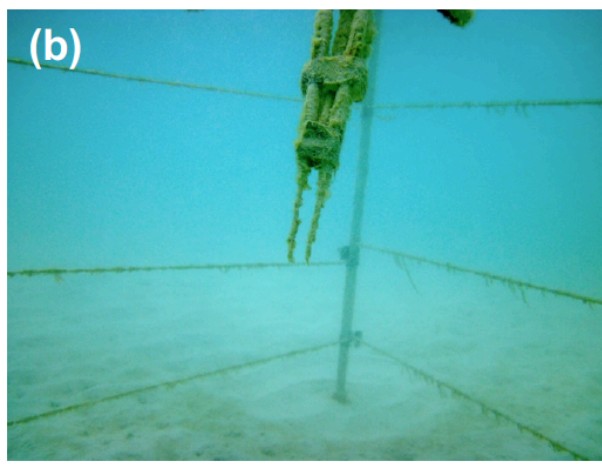

**Figure 1: (a) Microscopic image of marine snow particle showing organic and inorganic particles embedded in an gelatinous EPS matrix. (B) Close up of optode sensors with biofouling accumulated during a one week long deployment.**


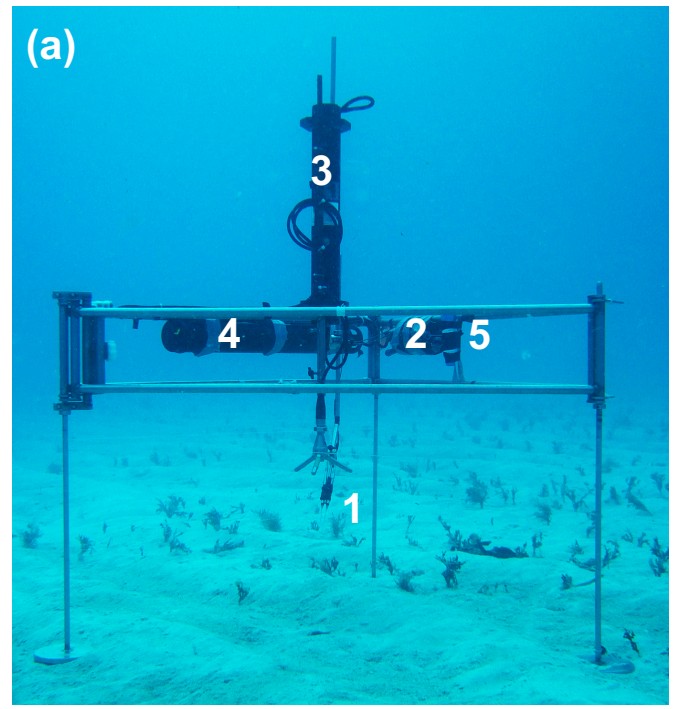
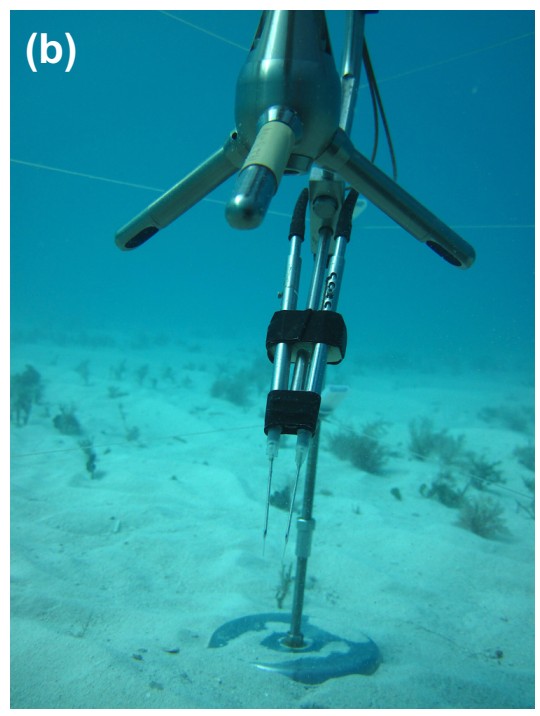


**Figure 2: (a) The eddy covariance instrument with dual fibre optode sensors at the study site. The tripod carries the optodes (1), underwater housing with oxygen meters reading the optodes (2), acoustic Doppler Velocimeter (ADV) (3) battery pack (4) and light logger (5) (b) Close up of the two optode sensors.**

Fig. 1. (A) The eddy covariance instrument with dual fiber optode sensor at the study site. The


tripod carries the optodes (1), underwater housing with oxygen meters reading the optodes (2),

acoustic Doppler Velocimeter (ADV) (3) battery pack (4) and light logger (5) (B) Close up of t

dual optode sensors.

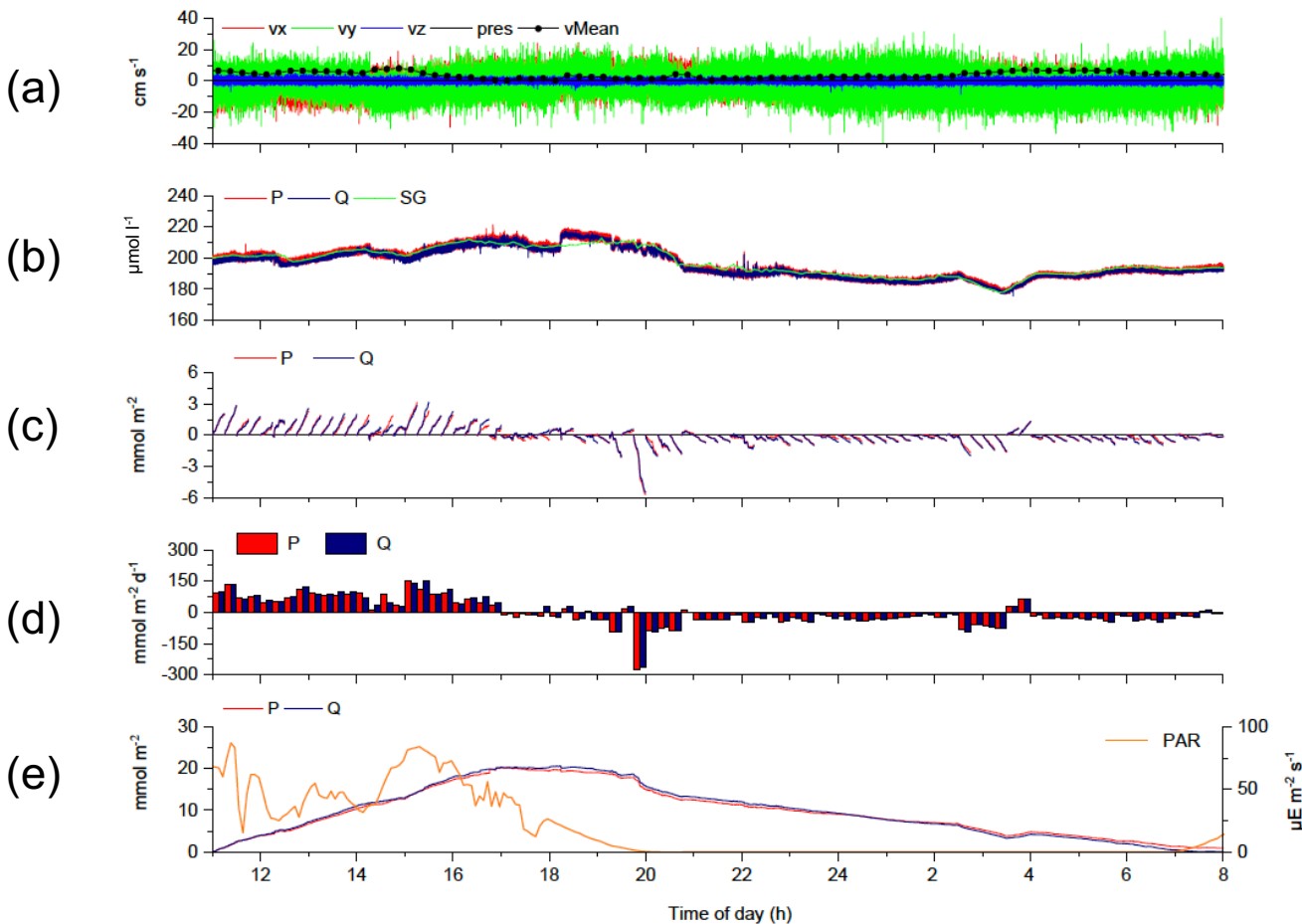

**Figure 3: (a) Case A, 2OEC deployment of 16-17 August 2013. A) horizontal x (red), y (green), and vertical z (blue) bottom flow components and mean current velocity (black circles). (b) Oxygen concentrations measured by the two optodes P (red) and Q (blue) and the Seaguard planar optode (green). (c) time-lag, rotation and storage corrected cumulative fluxes plotted for 15 min intervals. (d) average 15 min fluxes, (e) cumulative fluxes and PAR light intensity at the seafloor (orange line). Graphs produced with software Origin® 2017 (OriginLab).**


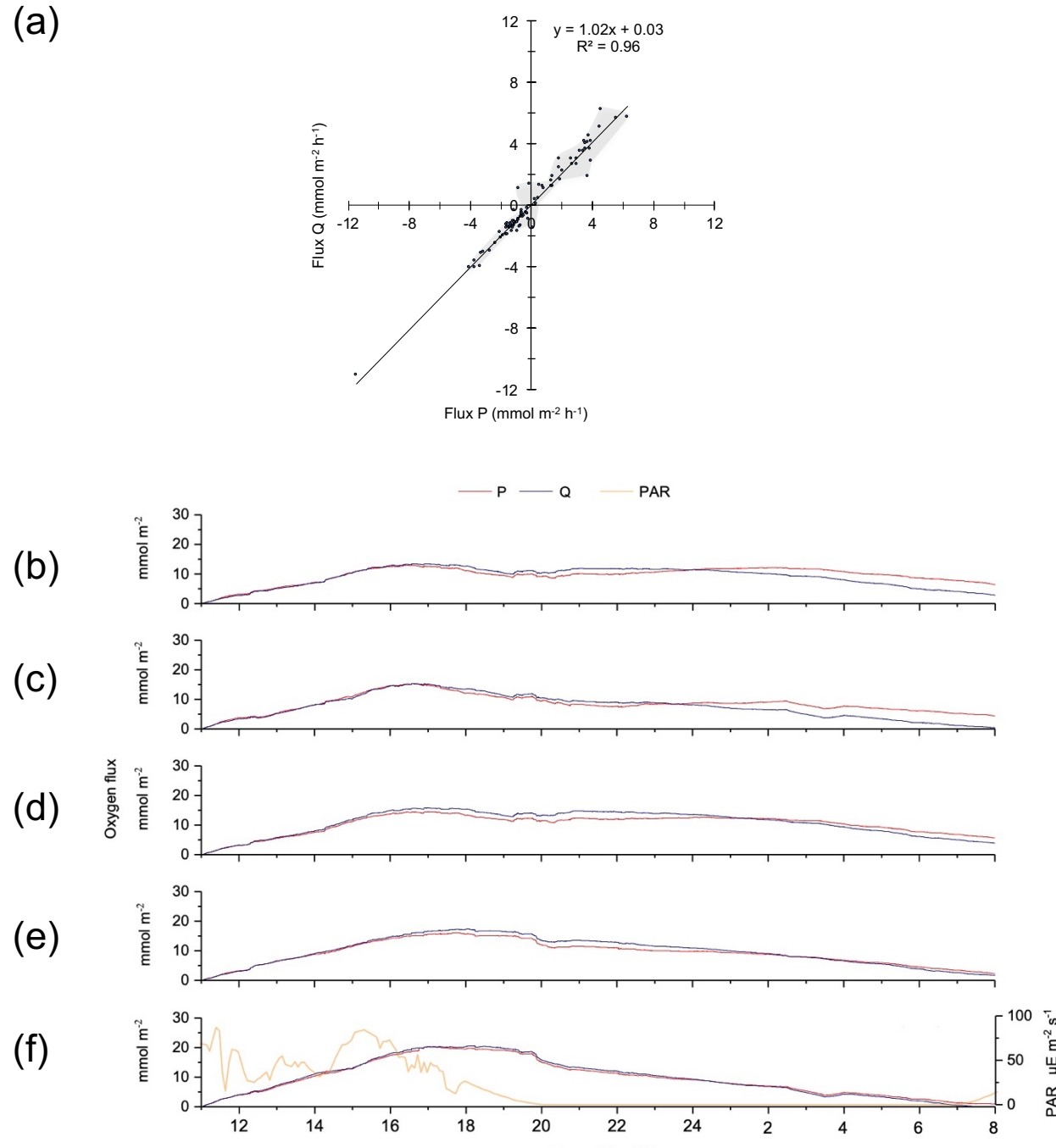

**Figure 4: (a) Comparison between the 15 minute fluxes based on optode P and Q signals for Case A. Grey shading depicts the envelope of the measured fluxes except the extreme value at -11, -11 mmol m$^{-2}$ h$^{-1}$. (b)-(f): Effects of the corrections applied to the flux calculations on cumulative flux. (b): cumulative fluxes calculated from the two optode signals without any correction (raw data). (c): Correction for temporal changes in the average water oxygen concentration. (d): Time lag correction, (e): Correction for wave rotation. (f): All corrections used in (c), (d) and (e) applied and the change in PAR light over time.**

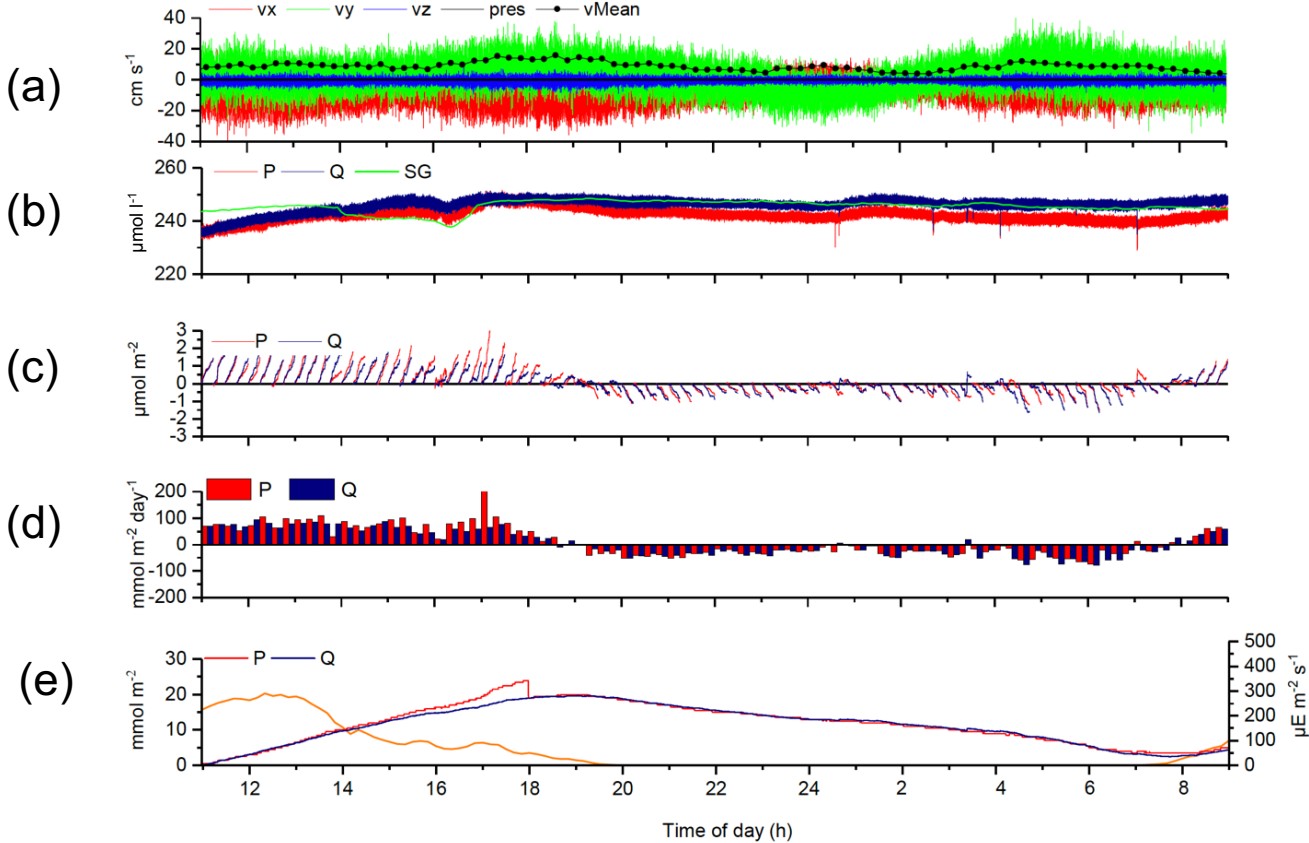

 **Figure 5: Case B, 2OEC deployment of 10-11 April 2014. Between 15:00 and 18:00, optode P was compromised (likely by marine snow attachment) and this phase was excluded from the calculations for average day and nighttime fluxes. P cumulative flux at 18:00 was intentionally reduced by 5 mmol m⁻² to allow comparison of the two cumulative fluxes based on P and Q data (Panel E). For further explanations of the panels see Fig. 3.**

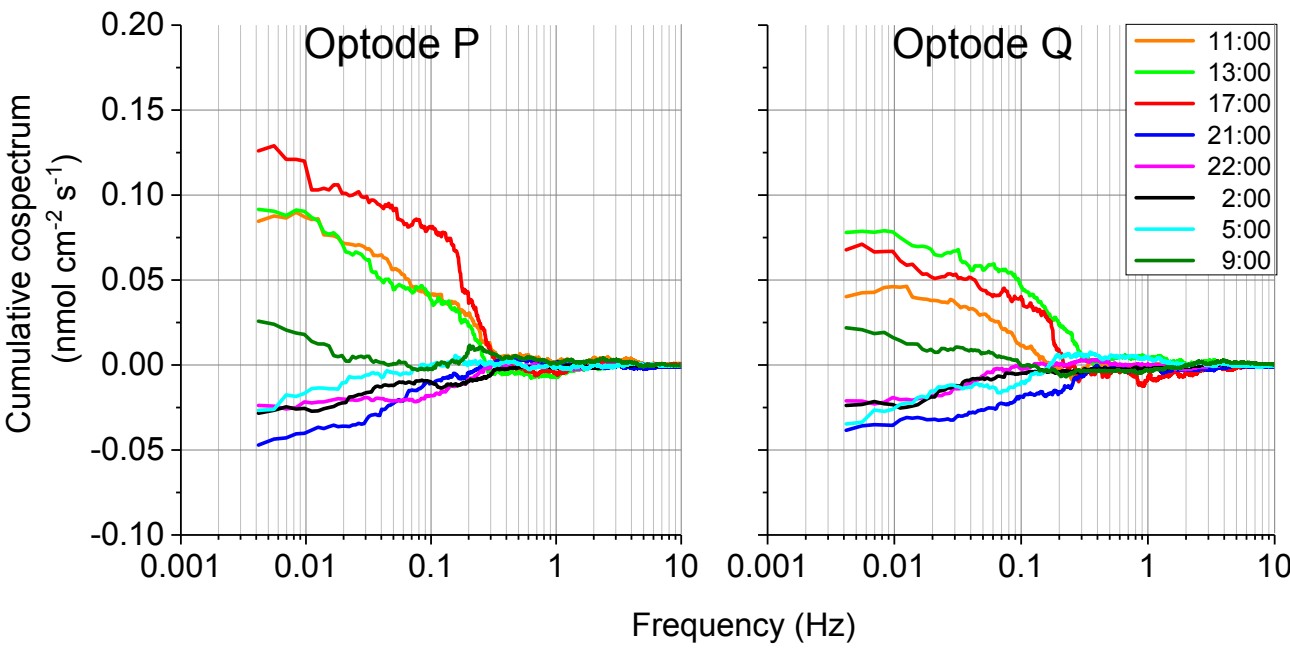

**Figure 6: Case B, 10-11 April 2014 2OEC deployment. Comparison of the cospectra for the two optodes P, Q at 17:00 revealed a steeper slope at the wave frequency 0.2-0.3 Hz. Cospectra processed using the SpectraVer1.2 software (P. Berg).**

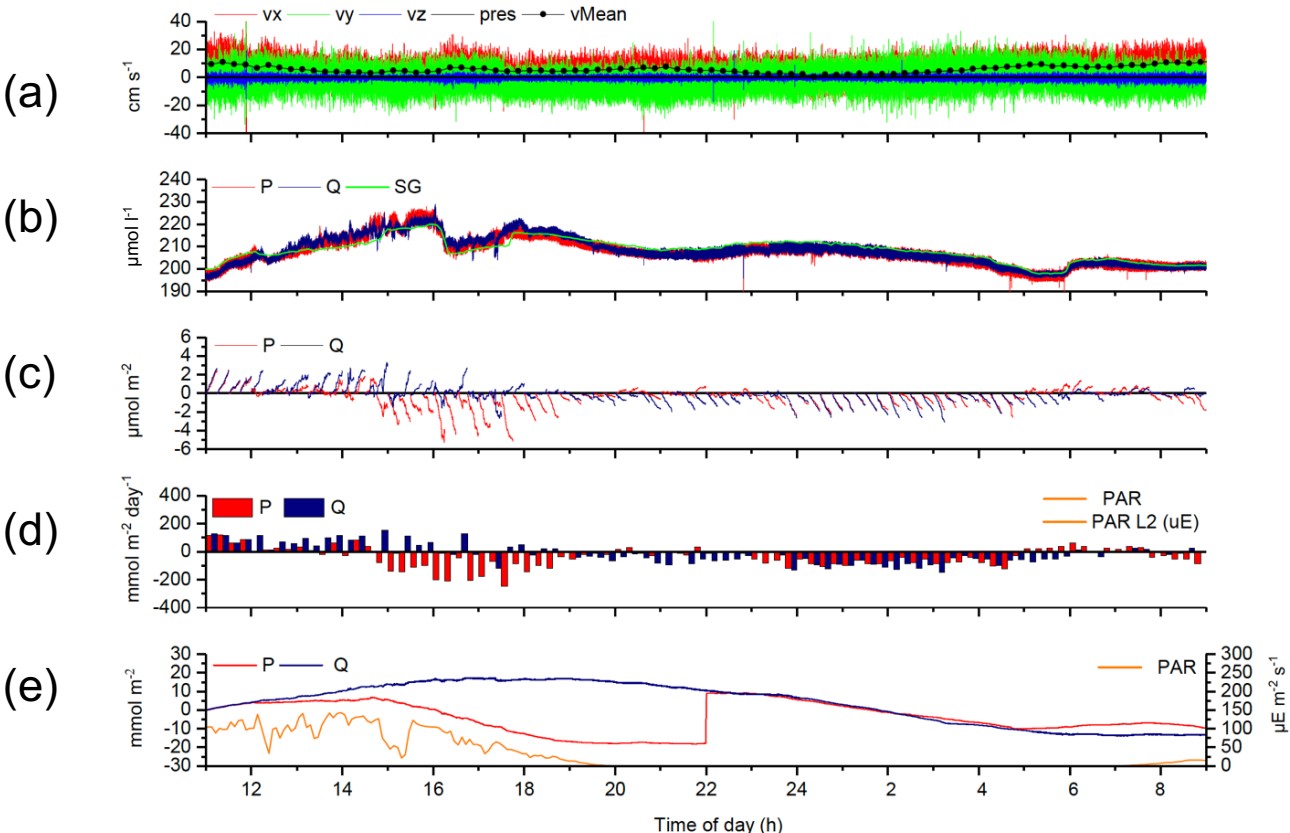

**Figure 7: Case C, 2OEC deployment of 14-15 August 2013. Between 12:00 and 22:00, optode P was compromised (likely by particle attachment) and again between 5:00 and 9:00 (loss of calibration). These phases were excluded from the calculations for average day and nighttime fluxes. P cumulative flux at 22:00 was intentionally increased by 27 mmol m$^{-2}$ to allow comparison of the two cumulative fluxes based on P and Q data (Panel E). For explanations of the panels see Fig. 3.**


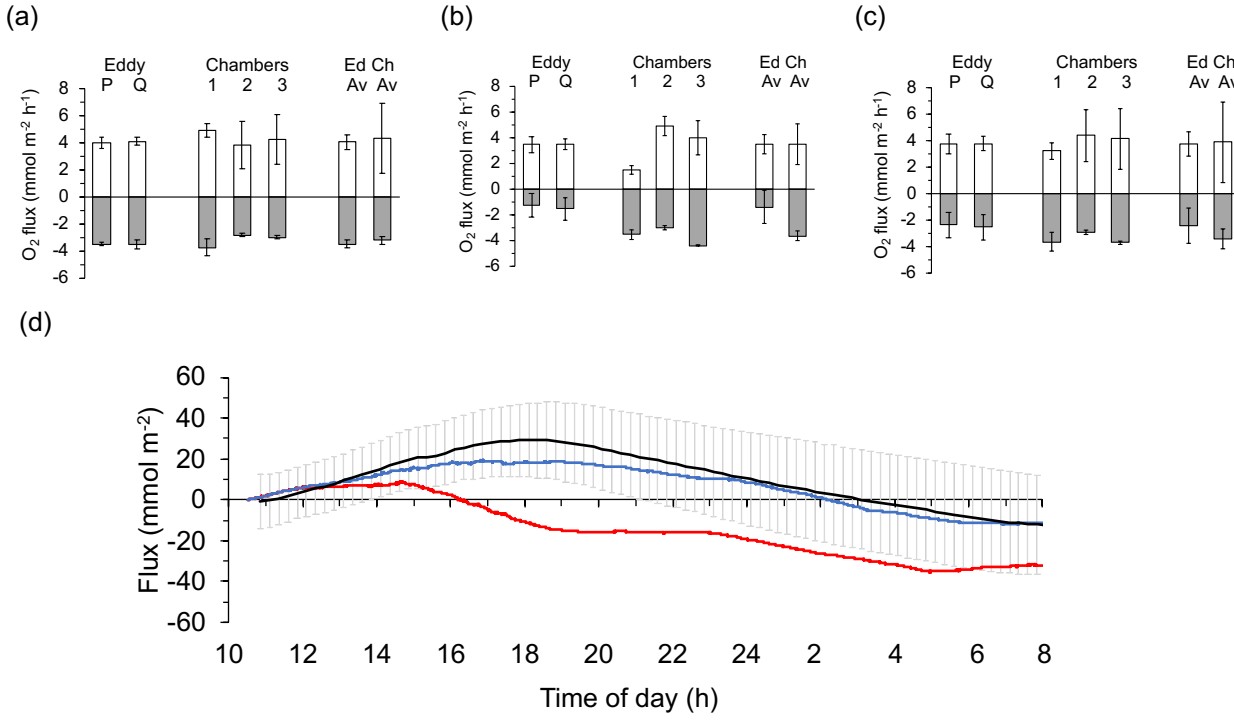

**Figure 8: Comparison of the day and nighttime fluxes recorded with the eddy covariance instrument and the benthic advection chambers. (a): 14-15 August deployments (Case C). (b): 16-17 August deployments (Case A). (c): Averages of the two August deployments. Light columns present daytime fluxes, dark columns nighttime fluxes. Columns on the right side of graphs (a), (b), and (c) labelled "Ed" depict the average of the fluxes based on sensors P and Q shown in the respective graph, columns labelled "Ch" depict the average of the fluxes recorded with the three chambers shown in the respective graph. Error bars represent standard deviation including error propagation. (d): Comparison of cumulative flux measured with the chambers (black line showing the average of the 3 chamber replicates and the standard deviation of the individual measuring points) and the 2OEC (red line: optode P, blue line: optode Q) during the 14-15 August deployments (Case C). The chamber fluxes confirmed that optode P was temporarily compromised during this deployment. Error bars depict standard deviation.**

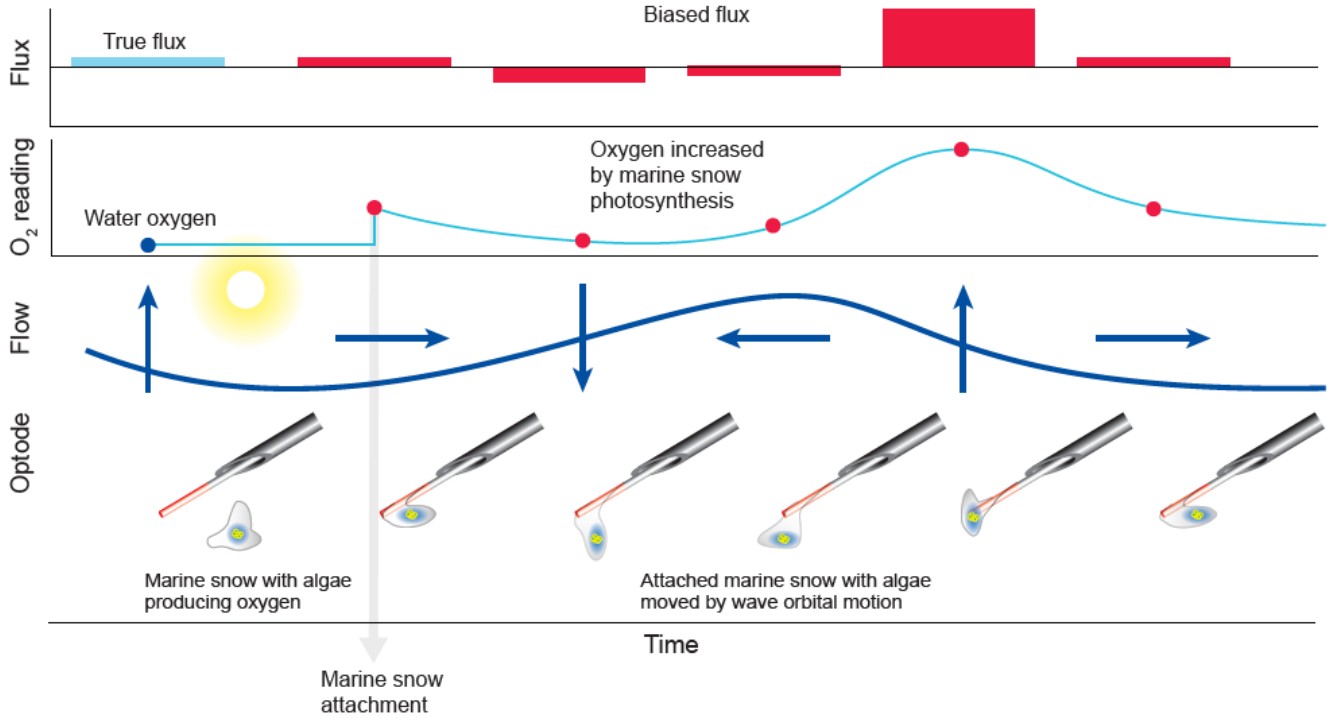

**Figure 9: False flux increase caused by the rhythmical deformation of a marine snow particle attached to an oxygen fibre optode. Erroneous fluxes result when wave orbital motion modulates the distance between photosynthesising organisms contained in the gelatinous marine snow particle and the sensing surface of the optode.**

685

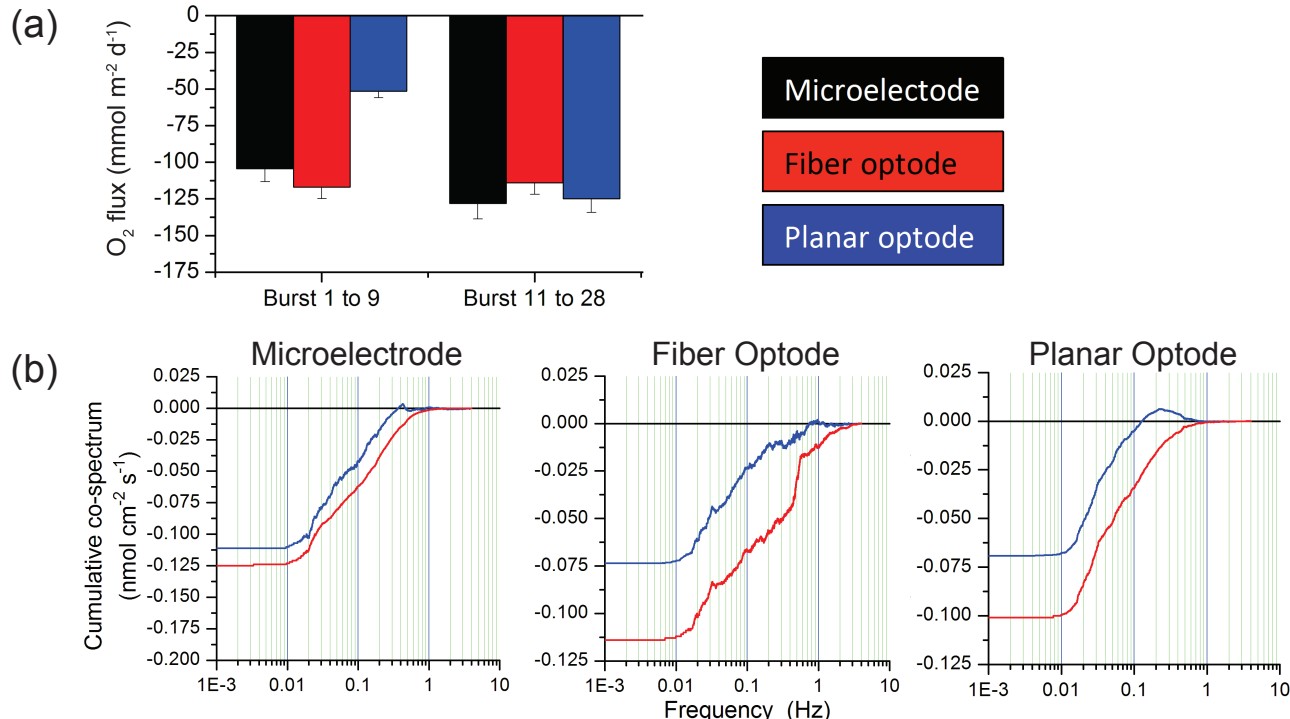

**Figure 10: Comparison of flux estimates generated by eddy covariance instruments equipped with either a Unisense microelectrode, a Pyroscience fibre optode or a Rinko planar optode. (a): comparison of time lag corrected fluxes for two burst intervals, each burst 15 minutes long. During burst 1-9, the sensors faced away from the flow, while in bursts 11 to 28, they faced the flow, (b): cumulative co-spectra of O₂ flux vs. frequency for the thee sensors. Blue lines represent the uncorrected data, red lines data after time lag correction.**

690