# Peer review of "Technical note: Measurements and data analysis of sediment-water oxygen flux using a new dual-optode eddy covariance instrument."

_Biogeosciences, 2020_

## Referee Comment (RC1) · Clare Reimers (Referee) · 2 Jun 2020

Review of "Technical note: Measurements and data analysis of sediment-water oxygen flux using a new dual-optode eddy covariance instrument."

The manuscript submitted by Huettel et al. is appropriate as a technical note because it focuses on issues related to the quality of oxygen sensor measurements in the context of aquatic eddy covariance (AEC) measurements of benthic oxygen fluxes. The authors stress biases that can occur when sensors are affected by biofouling, and they

illustrate with detailed examples how these artifacts can be recognized and controlled for using a dual-optode system. The examples are from an area of shallow shelf in the Florida Keys, making them unique environmentally. As a practitioner of these methods, I find this manuscript very useful, but I also recommend a number of revisions to improve clarity, especially for readers who may be less familiar with the AEC technique.

General recommendations: The manuscript is difficult to follow at times for reasons of organization and language. Most importantly, the introduction does not lead off with a very clear description of how biofouling or other "disturbances" can affect oxygen sensor measurements and corresponding AEC derivations. Instead the authors try to unravel these uncertain effects through the course of detailed reviews of data. More specific language throughout, as I will suggest below, would be helpful. Core questions are: does the biofouling produce or consume minute amounts of oxygen locally affecting what the sensor detects (sort of a contamination of the ambient condition), and why would this production or consumption be flow sensitive under waves? Zooming in to look at some data under both day and night conditions may help reveal the behavior.

It would also be helpful to simply refer to the three deployments used for illustration as something like "Case A, Case B and Case C". The dates of the deployments were so similar, that a reader has trouble differentiating the examples by date alone.

Specific suggestions for edits:

Figures 2, 4 and 6 panels (b) units should be micromoles per liter. (Use consistent unit designations in tables and figures).

There is duplication of references: McGinnis et al. 2008a and b are the same, Reimers et al. 2012a and b are the same.

Line by line:

Page 1 lines 9-10: First example of a vague reference to the core problem "but a main weakness of the commonly used instrumentation is the susceptibility of the delicate

oxygen microsensors required for the high frequency measurements to disturbances." This needs to be rewritten. Might be best to say something like "but a critical requirement is that EC sensors are able to resolve high frequency variations in dissolved oxygen concentration and vertical velocity without artifacts."

Page 1 lines 15-17. Revise. For example as: "Short-term changes in flux were confirmed or rejected with the 2OEC, giving more certain insights into the temporal dynamics of benthic oxygen flux in permeable carbonate sands."

Page 1 line 18. Why do you say "within a couple of hours"? Do you mean that this is how much time is needed to capture a representative flux under steady conditions?

Page 2 line 36. Add: Reimers et al., 2016. Microelectrode velocity effects and aquatic eddy covariance measurements under waves. J. Atm. Ocean. Tech. 33, 263-282.

Page 2. lines 40-42. I question the statements: "Optodes consume no oxygen and have very low or no stirring sensitivity (Holtappels et al., 2015). Compared to microelectrodes, they are less susceptible to signal drift and keep their calibration over longer time." It appears they may develop a stirring sensitivity once biofouled, and my experience is they may drift quite a bit due to their loss of sensitivity. Perhaps you could qualify these statements as: "Optodes consume no oxygen and may have very low or no stirring sensitivity (Holtappels et al., 2015). Compared to microelectrodes, we have observed they are less susceptible to signal drift and keep their calibration over longer time."

Page 2. lines 51-54. Here is where the authors need to give a clearer initial description of how biofouling will alter signals from an optical sensor. The statement "through shielding of the sensor tip from the water current and metabolic processes (i.e. respiration, photosynthesis)" is unclear. What kind of changes in signal magnitide and dynamics occur and why? These things are rarely "obvious", especially to new users.

Page 2. line 78. Revise as "is relatively robust compared to microelectrodes"...

[Figure]

Page 2. line 80. If the discussion of sensor drift and lifetime is based generally on previous measurements, make this clear. If it is based on the experiments in this paper, move this reporting to the results section.

Page 5. lines 136-137. Revise as: "the product of instantaneous oxygen fluctuation and instantaneous vertical velocity change" or something clearer.

Page 5. lines 140-146. The use of a storage term here is not well justified and later on is not clearly discussed. Is this the correction referred to in Figure 3C? Holtappels et al. (2013) illustrate transient contributions to eddy fluxes linked to changes in C, but their model predictions of these effects are different from the storage term (although both are dependent on dC/dt). At the heart of the matter is: does oxygen change due to advection or due to localized cumulative production of consumption in the bottom boundary layer? You appear to assume a changing diurnal "storage" balance in dissolved oxygen, but the oxygen time series show other drivers of change. The statement given at lines 202-204 also indicates you recognize advection.

Page 6. line 168. Here you start referring to data processing steps as "corrections". It would help the reader if section 2.3 separated these different corrections more clearly and let the reader know their effects on flux records would be evaluated as part of the results.

Page 6. line 172. It is not clear what the authors mean by "over the time course of the deployment". Can they indicate over what time intervals the cumulative slope was evaluated? Did they assess the slope burst by burst, or over longer intervals? How is the standard deviation derived for these calculations?

Page 7. lines 216-218. A better explanation of the signal produced by biofouling under waves needs to be given. I have seen this effect in my data too. An oscillation develops at the wave frequency that appears to be greater than what would occur if the water column gradient was moving up and down or back and forth with wave motions. Looking at segments of the oxygen, velocity and pressure time series may help sort

this out. It appears to be a "velocity effect".

Page 8. line 236. Here you discuss another reason for poor sensor performance (particle impact). This should also be mentioned in the introduction under optode weaknesses.

Page 9. line 286-287. State more specifically how current measurements can be affected and why. Differentiate between real changes in the flow reaching the ADV sampling volume (flow obstruction) and measurement artifacts due to acoustic returns off the sensor tip.

Page 10. The paper conclusions are relatively weak. The authors could easily expand a bit on how the fluxes measured in this study compare to other inner shelf and coastal environments with permeable sediments, e.g. those of Berg et al. 2013.

---

## Referee Comment (RC2) · Karl Attard (Referee) · 8 Jun 2020

**General comments**

Huettel et al. present a technical study describing a new dual-optode eddy covariance system. The authors integrate two independent O2 sensors within a standard eddy covariance setup to cross-check fluxes extracted using two independent O2 sensor output streams, and to identify any biases in the measurements which are most likely caused by sensor fouling. Dual O2 sensor eddy systems are not new per se (e.g.

McGinnis et al. 2011, Attard et al. 2014), but it is the first time that the two sensor signals have been compared in the level of detail provided in this study. The authors also perform chamber incubator measurements in parallel with eddy covariance to resolve O2 fluxes using two different state-of-the-art methods. Finally, the authors also provide a comparison between the three most popular O2 sensor systems for eddy covariance measurements.

The paper by Huettel et al will find broad interest among the growing community of aquatic eddy covariance users. The length of the paper and the angle of the study make it appropriate to be published as a Technical Note in Biogeosciences. The scientific methods are clearly outlined, language is fluent and precise, referencing is appropriate and up-to-date, and the overall presentation is well-structured and clear. I have one main comment and several smaller comments that the authors may wish to address. My main comment concerns how flux quality is evaluated. Currently, the authors determine quality based on (a) diel dynamics of O2 fluxes in relation to PAR, and (b) by comparison to chamber incubator measurements. If the quality-checking aspects could be expanded to include other metrics, then I foresee that the dual sensor approach would be useful in a broader range of settings.

Specific comments

**Introduction**

L20-67: It would be fair to mention that dual O2 sensor eddy systems have been in use for years (e.g. McGinnis et al. 2011 L&O Methods, Attard et al. 2014 L&O) but that so far, no detailed comparison between sensor signal output has been presented.

**Methods**

L70: It is worth mentioning that these meters only use half the potential voltage range of the Vector analog channels (0-2.5V), but developments are in place to increase this to the full range (0-5V).
L71: I cannot find this model on the Pyroscience website. Do you mean the FSO2-SUBPORT? https://www.pyroscience.com/en/products/all-meters/fso2-subport

L86-87: It would be useful to specify whether you powered the analog channels through the Vector

L95: Firesting O2Mini: Again here, please check that this is the right model, or specify whether this was an older model that has since been replaced by the FSO2-SUBPORT.

L97: 30mm is quite a large distance. Any reason for not moving closer to the measurement volume (e.g. 1 cm)?

L143-145: This reads like Results.

L148-161: It would be useful to describe the chamber measurements in more detail. How were the chamber O2 fluxes calculated? Did you have an optode inside the chamber measuring O2 concentration continuously? How long did the deployments last? Ultimately, what are we comparing in Fig. 7?

L156: Do the chambers attenuate PAR?

Results

L164: The first sentence seems out of place here.

L171-172: What deployment hours did you use for this analysis? Was it all of daylight hours i.e. until approx. 20:00? If so, does linear regression adequately represent these dynamics?

L179-183: I would expect that both optodes located at 35 cm above the seafloor and 1 cm apart would capture these variations, though?

L231: Fig 6E: I suppose that the jump at hour 22 in the cumulative flux for sensor P is not real, but it was offset in post processing to indicate that the two sensors match one another very well beyond this point. I understand the wish to illustrate this, but I think it
is confusing, because it suggests that despite the fluxes from both sensors being very different prior to hour 22, the daily integrated flux is very similar, which cannot be the case.

L233-234: I generally agree with this interpretation, it makes intuitive sense. One concern I have is that identifying what sensor works best at what time seems somewhat subjective. For instance, in Fig 6D hour 15, PAR drops from 150 to below 50 umol m-2 s-1, and sensor P registers a concurrent decrease in flux, but sensor Q does not. After hour 16 the fluxes from sensor P are clearly 'compromised', but then again, this assessment is based upon what we'd typically expect to see. I would otherwise be tempted to interpret the drop in fluxes at hour 15 in sensor P as 'real', unless there is some other metric we could use to establish flux quality.

Overall, I fear that if we do not adopt some quantitative metrics for establishing flux quality beyond what we expect to see (e.g. diel dynamics in relation to PAR), then we might miss out on something new and interesting. This is especially true during the nighttime or in non-photic habitats. In the absence of light, would we be able to say with the same certainty what flux dynamics is 'true' and what isn't?

We've been using a two-sensor setup since we started using eddy covariance in 2010, and I fully agree that this setup drastically increases the chances of obtaining good data. I typically evaluate the two sensor signals for their performance throughout the deployment by (a) comparing the mean O2 microsensor concentration to the O2 optode, (b) point-to-point noise in the 8 Hz data streams, and (c) linearity of the instantaneous cumulative fluxes for each 15 min flux period (Attard et al MEPS in press https://doi.org/10.3354/meps13372). Yamamoto et al (2015) L&O (https://doi.org/10.1002/lno.10018, Fig. 3) adopt a similar approach. Would fitting linear regressions to the cumulative instantaneous fluxes for each 15 min flux for sensors P and Q, and evaluating the coefficient of determination (R2 value), help to shed light on this? An additional analysis could be to fit P-I relationships to the data and see which sensor produces the best R2 value, like the approach described in Attard & Glud
(2020) Biogeosciences Discussions (https://doi.org/10.5194/bg-2020-140, Fig. 2).

L243-244: Also here, it would be good to mention what part of the integrated curve was used for this analysis.

**Discussion**

L222-271: A two sensor setup provides redundancy and cross comparison, no question about that. However, it is also twice the cost in hardware, and twice the amount of work in postprocessing. If fouling seems to be such an issue, wouldn't the right approach be to try to eliminate fouling, rather than to add more sensors? I believe there is scope in the Discussion and in the Conclusion to comment on what future modifications might be valuable. For instance, should we install a pump and back-flush the sensors before each measurement burst? Can we monitor buildup of sensor fouling in some other way?

**Technical corrections**

- L78: Remove extra 'relative'
- L81: ...established using the jet-nozzle method...
- L86: Analogue should read 'analog'
- L127: Apostrophes should be replaced with primes
- L137: should read 'products of instantaneous...'
- L156: 'permitting' rather than 'facilitating'
- L174: Should read 'The close agreement...'

Please also note the supplement to this comment: https://www.biogeosciences-discuss.net/bg-2020-172/bg-2020-172-RC2supplement.pdf Interactive comment

---

## Author Comment (AC1) · 26 Jun 2020

Response to reviewers comments and questions
Review of "Technical note: Measurements and data analysis of sediment-water oxygen

flux using a new dual-optode eddy covariance instrument."

The manuscript submitted by Huettel et al. is appropriate as a technical note because it focuses on issues related to the quality of oxygen sensor measurements in the context of aquatic eddy covariance (AEC) measurements of benthic oxygen fluxes. The authors stress biases that can occur when sensors are affected by biofouling, and they illustrate with detailed examples how these artifacts can be recognized and controlled for using a dual-optode system. The examples are from an area of shallow shelf in the Florida Keys, making them unique environmentally. As a practitioner of these methods, I find this manuscript very useful, but I also recommend a number of revisions to improve clarity, especially for readers who may be less familiar with the AEC technique.

Response: We thank Dr. Reimers for the detailed review of our manuscript and the helpful comments and questions.

General recommendations: The manuscript is difficult to follow at times for reasons of organization and language. Most importantly, the introduction does not lead off with a very clear description of how biofouling or other "disturbances" can affect oxygen sensor measurements and corresponding AEC derivations. Instead the authors try to unravel these uncertain effects through the course of detailed reviews of data.

Response: We added a paragraph explaining description how disturbances including biofouling can affect measurements and corresponding AEC derivations. P2L53: "Irrespective of the technology, the readings of the oxygen sensors can be biased by attachment of particles, bacteria or algal cells, which can affect the sensor signal through shielding of the sensor tip and metabolic processes (Smith et al., 2007;Delauney et al., 2010). Mineral particles may be impenetrable to gases, while organic particles may be sufficiently dense or oxygen consuming such that oxygen diffusion through them is reduced, thereby decreasing and delaying oxygen transport to the sensing surface (Zetsche et al.;Ploug and Passow, 2007). The ensuing increase in the response time of the sensor dampens the oxygen signal and thereby reduces the calculated flux. The

most common particles attaching to sensors may be marine snow particles (Fig. 1 a), sticky aggregates of various organic and inorganic particles glued together by extra-cellular polymeric substances (Alldredge and Silver, 1988). Bacteria and phytoplank-ton cells commonly contained in these particles can cause oxygen consumption and oxygen production, thereby affecting the signals of the oxygen sensor and the fluxes calculated from these readings."

We also added a sentence and figure explaining how a marine snow particle attached to the oxygen sensor can lead to increased flux estimates when waves are present.

P9L308: "A marine snow particle with photosynthesizing organisms attached to the tip of the oxygen sensor P may have caused the erroneous flux estimates. Oxygen con-centration in the centre of such aggregates during light conditions can be increased by 85 % relative to the surrounding water (Ploug and Jorgensen, 1999), or even by 180% within millimetre-size gelatinous colonies of Phaeocystis spp., a common global bloom-forming phytoplankton organism (Ploug et al., 1999). The movement of such an attached photosynthesizing particle by wave orbital motion can synchronize vertical current flow oscillations and the effect of the particle on the oxygen reading (e.g. in-creased oxygen due to photosynthesis) and thereby lead to erroneous flux estimates (Fig. X.)"

More specific language throughout, as I will suggest below, would be helpful. Core questions are: does the biofouling produce or consume minute amounts of oxygen locally affecting what the sensor detects (sort of a contamination of the ambient con-dition), and why would this production or consumption be flow sensitive under waves? Zooming in to look at some data under both day and night conditions may help reveal the behavior.

Response: If one of the two parallel measuring sensors showed a temporary increase or drop in oxygen as found in the deployments on 10-11 April 2014 and 14-15 August 2013, we attributed this to the biofouling of that sensor, and in-situ inspections of the

sensors revealed biofouling (extreme case now shown in Fig. 1b). We inspected the data and as an example provide the co-spectra shown in Figure 6 that reveal a temporary sensitivity to waves in sensor P, which we explain with the process now depicted in Fig. 10. Measurements have shown that marine snow particles can produce and consume substantial amounts of oxygen (see references listed in the response above) and marine snow was abundant at the study site partly due to the proximity of the coral reefs that release mucus to the water. We added this explanation to the text: P10L304 "If one of the two parallel measuring sensors showed a temporary increase or drop in oxygen as found in the deployments on 10-11 April 2014 and 14-15 August 2013, we attributed this to the biofouling of that sensor, and in-situ inspections of the sensors revealed biofouling (extreme case now shown in Fig. 1b). Marine snow was abundant at the study site partly due to its proximity to coral reefs that release mucus to the water (Wild et al., 2004).

and Figure 9: False flux increase caused by the rhythmical deformation of a marine snow particle attached to an oxygen fibre optode. Erroneous fluxes result when wave orbital motion modulates the distance between photosynthesising organisms contained in the gelatinous marine snow particle and the sensing surface of the optode.

It would also be helpful to simply refer to the three deployments used for illustration as something like "Case A, Case B and Case C". The dates of the deployments were so similar, that a reader has trouble differentiating the examples by date alone.

Response: We followed the suggestion of the reviewer and now use "Case A, Case B and Case C".

Specific suggestions for edits: Figures 2, 4 and 6 panels (b) units should be micromoles per liter. (Use consistent unit designations in tables and figures).

Response: Done.

There is duplication of references: McGinnis et al. 2008a and b are the same, Reimers

et al. 2012a and b are the same.

Response: Thanks for pointing this out, we removed the duplication

Line by line: Page 1 lines 9-10: First example of a vague reference to the core problem "but a main weakness of the commonly used instrumentation is the susceptibility of the delicate oxygen microsensors required for the high frequency measurements to disturbances." This needs to be rewritten. Might be best to say something like "but a critical requirement is that EC sensors are able to resolve high frequency variations in dissolved oxygen concentration and vertical velocity without artifacts."

Response: We followed the suggestion of the reviewer and changed the sentence. It now reads: P1L8 In-situ fluxes can be measured non-invasively with the aquatic eddy covariance technique, but a critical requirement is that the sensors of the instrument are able to correctly capture the high frequency variations in dissolved oxygen concentration and vertical velocity".

Page 1 lines 15-17. Revise. For example as: "Short-term changes in flux were confirmed or rejected with the 2OEC, giving more certain insights into the temporal dynamics of benthic oxygen flux in permeable carbonate sands."

Response: We revised the sentence that now reads: P1L15 "Short-term changes in flux that are unsupported in measurements with single oxygen sensor instruments can be confirmed or rejected with the 2OEC and in our deployments provided new insights into the temporal dynamics of benthic oxygen flux in permeable carbonate sands."

Page 1 line 18. Why do you say "within a couple of hours"? Do you mean that this is how much time is needed to capture a representative flux under steady conditions?

Response: We clarified our statement following the suggestion of the reviewer. It now reads: P1L17 "Under steady conditions, representative benthic flux data can be generated with the 2OEC within a couple of hours, making this technique suitable for mapping sediment-water, intra-water column, or atmosphere-water fluxes".

[Figure]

Page 2 line 36. Add: Reimers et al., 2016. Microelectrode velocity effects and aquatic eddy covariance measurements under waves. J. Atm. Ocean. Tech. 33, 263-282.

Response: Done.

Page 2. lines 40-42. I question the statements: "Optodes consume no oxygen and have very low or no stirring sensitivity (Holtappels et al., 2015). Compared to microelectrodes, they are less susceptible to signal drift and keep their calibration over longer time." It appears they may develop a stirring sensitivity once biofouled, and my experience is they may drift quite a bit due to their loss of sensitivity. Perhaps you could qualify these statements as: "Optodes consume no oxygen and may have very low or no stirring sensitivity (Holtappels et al., 2015). Compared to microelectrodes, we have observed they are less susceptible to signal drift and keep their calibration over longer time."

Response: We followed the recommendation of the reviewer, and the sentence now reads: P2L41 "Optodes consume no oxygen and may have very low or no stirring sensitivity (Holtappels et al., 2015). Compared to microelectrodes, we have observed they are less susceptible to signal drift and keep their calibration over longer time."

Page 2. lines 51-54. Here is where the authors need to give a clearer initial description of how biofouling will alter signals from an optical sensor. The statement "through shielding of the sensor tip from the water current and metabolic processes (i.e. respiration, photosynthesis)" is unclear. What kind of changes in signal magnitide and dynamics occur and why? These things are rarely "obvious", especially to new users.

Response: Thank you for pointing this out. We added the following information: P2L53 "Irrespective of the technology, the readings of the oxygen sensors can be biased by attachment of particles, bacteria or algal cells, which can affect the sensor signal through shielding of the sensor tip and metabolic processes (Smith et al., 2007;Delauney et al., 2010). Mineral particles may be impenetrable to gases, while organic particles may be sufficiently dense or oxygen consuming such that oxygen diffusion through them

is reduced, (Zetsche et al.;Ploug and Passow, 2007) thereby decreasing and delaying oxygen transport to the sensing surface. The ensuing increase in the response time of the sensor dampens the oxygen signal and thereby reduces the calculated flux. Berg et al. (2015) explained how a time offset between the oxygen and the velocity data can cause significant over- or underestimation of the flux. The most common particles attaching to sensors may be marine snow particles (Fig. 1 a), sticky aggregates of various organic and inorganic particles glued together by extracellular polymeric substances (Alldredge and Silver, 1988). Bacteria and phytoplankton cells commonly contained in these particles can cause oxygen consumption and oxygen production, thereby affecting the signals of the oxygen sensor and the fluxes calculated from these readings. We observed oxygen flux increases up to 4.4 mmol m-2 h-1 caused by photosynthesis and decreases up to -5.2 mmol m-2 h-1 caused by respiration of microbes contained in marine snow attached to the oxygen sensor."

Page 2. line 78. Revise as "is relatively robust compared to microelectrodes". . .

Response: We followed the suggestion of the reviewer, and the sentence now reads: P3L73 "With the advantages of being relatively robust compared to microelectrodes and less expensive, optodes are predisposed to become the preferable sensor-type for aquatic eddy covariance measurements"

Page 2. line 80. If the discussion of sensor drift and lifetime is based generally on previous measurements, make this clear. If it is based on the experiments in this paper, move this reporting to the results section.

Response: Sensor lifetime and drift were observed in previous field deployments. We added this information to the text: P3L93 "Our previous field measurements indicated that when operated continuously at a measuring frequency of $\sim$ 8 Hz, the useful lifetime of the OXR430-UHS typically was 3 to 7 days before the signal decreased to a level precluding reliable data interpretation. The signal drift over this period was negligible (< 0.03%) (Huettel, unpublished).

Page 5. lines 136-137. Revise as: "the product of instantaneous oxygen fluctuation and instantaneous vertical velocity change" or something clearer.

Response: Done

Page 5. lines 140-146. The use of a storage term here is not well justified and later on is not clearly discussed. Is this the correction referred to in Figure 3C? Holtappels et al. (2013) illustrate transient contributions to eddy fluxes linked to changes in C, but their model predictions of these effects are different from the storage term (although both are dependent on dC/dt). At the heart of the matter is: does oxygen change due to advection or due to localized cumulative production of consumption in the bottom boundary layer? You appear to assume a changing diurnal "storage" balance in dissolved oxygen, but the oxygen time series show other drivers of change. The statement given at lines 202-204 also indicates you recognize advection.

Response: We agree with the reviewer that this was not explained sufficiently. We added the following text: P5L155 "At our measuring height of 35 cm above the seafloor, the diurnal fluctuation in mean water column oxygen concentration can result in substantial changes in the oxygen inventory of the water column below the measuring volume, which can bias the local eddy flux measurements. To correct for this effect, an oxygen storage term, calculated as $\int_0^h dC/dt\, h$, was subtracted from the measured eddy flux to determine the benthic oxygen flux ($dC/dt$ = change of the average oxygen concentration over time, calculated through linear detrending of the measured oxygen data over 15 height of the measuring volume) (Rheuban et al., 2014a).

Page 6. line 168. Here you start referring to data processing steps as "corrections". It would help the reader if section 2.3 separated these different corrections more clearly and let the reader know their effects on flux records would be evaluated as part of the results.

Response: We agree with the reviewer and moved the effects of the flux corrections we applied to the results section.

[Figure]

Page 6. line 172. It is not clear what the authors mean by "over the time course of the deployment". Can they indicate over what time intervals the cumulative slope was evaluated? Did they assess the slope burst by burst, or over longer intervals? How is the standard deviation derived for these calculations?

Response: We added the following text to clarify this point: P5L163 "For the comparison of the temporal evolution of the fluxes that were determined using the recordings of the two optodes, we calculated the cumulative fluxes over the duration of the deployments. The slopes of the increasing cumulative fluxes during daylight and decreasing cumulative fluxes during nighttime were assessed for hourly time intervals, and standard deviations of the fluxes reflect the deviations between three hourly slope determinations.

Page 7. lines 216-218. A better explanation of the signal produced by biofouling under waves needs to be given. I have seen this effect in my data too. An oscillation develops at the wave frequency that appears to be greater than what would occur if the water column gradient was moving up and down or back and forth with wave motions. Looking at segments of the oxygen, velocity and pressure time series may help sort this out. It appears to be a "velocity effect".

Response: We added an explanation and figure 9 P10L308 "A marine snow particle with photosynthesizing organisms attached to the tip of the oxygen sensor P may have caused the erroneous flux estimates. Oxygen concentration in the centre of such aggregates during light conditions can be increased by 85 % relative to the surrounding water (Ploug and Jorgensen, 1999), or even by 180% within millimetre-size gelatinous colonies of Phaeocystis spp., a common global bloom-forming phytoplankton organism (Ploug et al., 1999). The movement of such an attached photosynthesizing particle by wave orbital motion can synchronize vertical current flow oscillations and the effect of the particle on the oxygen reading (e.g. increased oxygen due to photosynthesis) and thereby lead to erroneous flux estimates (Fig. 10.)

Page 8. line 236. Here you discuss another reason for poor sensor performance (particle impact). This should also be mentioned in the introduction under optode weaknesses.

Response: We added the following sentence in the introduction P2L46 "Although most optodes are more robust than microelectrodes, they can break due to particle collision."

Page 9. line 286-287. State more specifically how current measurements can be affected and why. Differentiate between real changes in the flow reaching the ADV sampling volume (flow obstruction) and measurement artifacts due to acoustic returns off the sensor tip.

Response: We added the following explanation: P11L324 "A cylindrical sensor placed in the path of the flow upstream the ADV measuring volume can shed a vortex street thereby compromising the flow in the measuring volume and the flux estimates based on the flow measurements. Depending on the flow Reynolds number, such vortices may extend between 5 to 20 times the diameter of the cylinder downstream the sensor (Green, 2012). By using the Pyroscience fiber optode for the 2OEC, one of the smallest and fastest oxygen sensors presently available, potential errors caused by the disturbance of the flow and interference with the acoustic pulses of the Doppler velocimeter can be avoided. At the turbulent Reynolds numbers typical for our study site (4000 < Re < 110000), the vortices shed by the 430 $\mu$m fiber exposed to the water currents extend between 2 to 10 mm downstream of the fiber (Green, 2012). Since the tips were placed at 30 mm horizontal distance from the lower edge of the ADV measuring volume, turbulence caused by fiber-flow interaction could not reach the ADV measuring volume. Similarly, the sensor tips at that distance did not interfere with the acoustic pulses of the ADV, and when initially positioning the optode tips, we confirmed that the optode fibers did not cause any disturbances in the ADV signal."

Page 10. The paper conclusions are relatively weak. The authors could easily expand a bit on how the fluxes measured in this study compare to other inner shelf and coastal
environments with permeable sediments, e.g. those of Berg et al. 2013.

Response: We agree with the reviewer that the discussion of the flux results could be expanded, however, this paper was designed to introduce the instrument and the data interpretation, and, with all due respect, decided not to expand the discussion of the flux results in this paper. We are presently working on a manuscript that uses the results from these deployments together with other flux data measured at this study site to demonstrate the high metabolic activity of the coarse carbonate sands and to discuss their role in the coral reef ecosystem. This paper will also include a comparison of the fluxes presented here with fluxes measured in other inner shelf environments.

---

## Author Comment (AC2) · 26 Jun 2020

General comments Huettel et al. present a technical study describing a new dualoptode eddy covariance system. The authors integrate two independent O2 sensors within a standard eddy covariance setup to cross-check fluxes extracted using two independent O2 sensor output streams, and to identify any biases in the measurements which are most likely caused by sensor fouling. Dual O2 sensor eddy systems are not new per se (e.g. McGinnis et al. 2011, Attard et al. 2014), but it is the first time that the two sensor signals have been compared in the level of detail provided in this study. The authors also perform chamber incubator measurements in parallel with eddy covariance to resolve O2 fluxes using two different state-of-the-art methods. Finally, the authors also provide a comparison between the three most popular O2 sensor systems for eddy covariance measurements. The paper by Huettel et al will find broad interest among the growing community of aquatic eddy covariance users. The length of the paper and the angle of the study make it appropriate to be published as a Technical Note in Biogeosciences. The scientific methods are clearly outlined, language is fluent and precise, referencing is appropriate and up-to-date, and the overall presentation is well-structured and clear. I have one main comment and several smaller comments that the authors may wish to address.

Response: We thank Dr. Attard for the detailed review of our manuscript and the helpful comments and questions.

My main comment concerns how flux quality is evaluated. Currently, the authors determine quality based on (a) diel dynamics of O2 fluxes in relation to PAR, and (b) by comparison to chamber incubator measurements. If the quality-checking aspects could be expanded to include other metrics, then I foresee that the dual sensor approach would be useful in a broader range of settings.

Response: We agree with the reviewer that assessing the quality, validity and accuracy of the fluxes is central when conducting non-invasive eddy covariance measurements. In addition to the two methods mentioned by the reviewer (i.e. diel flux dynamics in relation to PAR, and comparison to chamber measurements), we use here the agreement/disagreement between the fluxes calculated from the signals of two indepen-
dently measuring optodes as a tool to assess the quality of the measured fluxes. To make that point more clear, we added the following text to the conclusions: P11L350 "We propose using the agreement/disagreement between the fluxes calculated from the signals of two independently measuring optodes as a tool to assess the quality of the measured fluxes. The nearly identical cumulative fluxes calculated from the two optodes in our August (Case A) and April (Case B) deployments strongly imply that the dynamics of the fluxes were measured accurately by the system. Likewise, the linearity of the cumulative flux increase during daytime and decrease during nighttime (Figs. 3e, 5e) and the very similar slopes of these cumulative flux curves support that the measurements recorded representative fluxes. The good agreement of the fluxes measured with the eddy covariance instrument and the fluxes measured with independently with a very different method (advection chambers, Fig. 8a, b, d) indicate that the magnitudes of the fluxes recorded by the 2OEC are correct."

Specific comments Introduction L20-67: It would be fair to mention that dual O2 sensor eddy systems have been in use for years (e.g. McGinnis et al. 2011 L&O Methods, Attard et al. 2014 L&O) but that so far, no detailed comparison between sensor signal output has been presented.

Response: We thank the reviewer for pointing out the missing references to other dual sensor instruments. In a earlier version of the manuscript, we had an extended discussion of microlectrode-equipped eddy instruments, which included the dual electrode systems deployed by Attard et al (2014) and McGinnis et al. (2011). This matter was removed when shortening the paper and we now re-inserted these references. To the best of our knowledge, eddy instruments based on dual optode measurements have not been introduced, and we are not aware of publications that use the date evaluation approach explained here. The following section was inserted into the text: P2L35 "To improve the reliability of the flux measurements, eddy covariance with dual microelectrodes were developed, e.g. (Attard et al., 2014;McGinnis et al., 2011;Rodil et al., 2019;de Froe et al., 2019;Rovelli et al., 2015)"
Methods L70: It is worth mentioning that these meters only use half the potential voltage range of the Vector analog channels (0-2.5V), but developments are in place to increase this to the full range (0-5V).

Response: We thank the reviewer for this suggestion and added the following sentence to Table A1 in Appendix A: "the meter presently uses half the potential voltage range of the Vector analog input, but developments are in place to increase this to the full range (0-5V)"

L71: I cannot find this model on the Pyroscience website. Do you mean the FSO2-SUBPORT? https://www.pyroscience.com/en/products/all-meters/fso2-subport

Response: We thank the reviewer for pointing this out. The Pyroscience TM FireStingO2-Mini oxygen meter recently has been replaced by the Pyroscience TM PICO-O2, which is similar to the meter we used but more compact. The FireStingO2-Mini oxygen meter is still available in combination with the Pyroscience TM FSO2-SUBPORT. We added the following text to the legend of Table A1 in Appendix A:

This oxygen meter recently has been replaced by the PyroscienceTM PICO-O2, which is similar to the FireStingO2-Mini but more compact. The FireStingO2-Mini oxygen meter is still available in combination with the PyroscienceTM subport (FSO2-SUBPORT).

L86-87: It would be useful to specify whether you powered the analog channels through the Vector

Response: Thank you for this suggestion. We added P3L86 "The FireSting O2-Mini oxygen meters were supplied with the output power of the ADV (see below)."

L95: Firesting O2Mini: Again here, please check that this is the right model, or specify whether this was an older model that has since been replaced by the FSO2-SUBPORT.

Response: To avoid confusion we added the following text: P3L84 "(specifications listed in Table A1 in Appendix A, now sold in combination with the PyroscienceTM sub-
port (FSO2-SUBPORT)"

L97: 30 mm is quite a large distance. Any reason for not moving closer to the measurement volume (e.g. 1 cm)?

Response: We chose this distance to prevent any potential interference with the flow and acoustic pulses of the Vector ADV. We added the following sentence to make this clear P4L114 "This distance prevents any disturbance of the flow in the measuring volume of the ADV and any interference with the acoustic pulses of the Vector."

L143-145: This reads like Results.

Response: We agree and moved this sentence to the results section.

L148-161: It would be useful to describe the chamber measurements in more detail. How were the chamber O2 fluxes calculated? Did you have an optode inside the chamber measuring O2 concentration continuously? How long did the deployments last? Ultimately, what are we comparing in Fig. 7?

Response: We added the following paragraph providing more detail about the chamber measurements: P6L179 "The acrylic cylinder of the chambers was pushed 12 cm into the sand sediment, resulting in a chamber water volume of 5 L. A Hach-Rigid-O2-Optode mounted in the chamber lid collected oxygen concentrations at 15 minute time intervals. The fluxes were calculated from the changes of the oxygen concentration in the water column of the chamber over time. Chamber incubations ran for 24 h, then the lid was opened to allow re-equilibration with the ambient water before starting the next measurement cycle."

L156: Do the chambers attenuate PAR? Results

Response: Thank you for asking this question. We measured a 10% loss in PAR caused by the acrylic chamber lids and added this information to the text P6L178 "(10% loss in PAR through light attenuation caused by the acrylic)"

BGD
L164: The first sentence seems out of place here.

Response: We structured the Results sections by these subtitles, and the purpose of these subtitles becomes more clear when reading the related paragraphs.

L171-172: What deployment hours did you use for this analysis? Was it all of daylight hours i.e. until approx. 20:00? If so, does linear regression adequately represent these dynamics?

Response: We used all daylight hours and the flux increase during daylight could be represented well by linear regression (R2 > 0.9). The same applied to the nighttime fluxes. When calculating the fluxes, we avoided sections with disturbances in the cumulative flux curves. We added the following explanation to the text: P7L199 "The flux increase during daylight and decrease during nighttime could be represented well by linear regression (R2 > 0.9)."

L179-183: I would expect that both optodes located at 35 cm above the seafloor and 1 cm apart would capture these variations, though?

Response: They do capture these variations, however, the higher heterogeneity of the oxygen distribution in the water still causes larger differences between the simultaneous readings of the two optodes during daylight hours (Fig. 4a)

L231: Fig 6E: I suppose that the jump at hour 22 in the cumulative flux for sensor P is not real, but it was offset in post processing to indicate that the two sensors match one another very well beyond this point. I understand the wish to illustrate this, but I think it is confusing, because it suggests that despite the fluxes from both sensors being very different prior to hour 22, the daily integrated flux is very similar, which cannot be the case.

Response: With all due respect, we disagree with the reviewer here. As pointed out in the legend of the figure, P cumulative flux at 18:00 was intentionally reduced by 5 mmol m-2 to allow comparison of the two cumulative fluxes based on P and Q data
(Panel E). After excluding the time period during which sensor P was compromised by biofouling, the daily integrated fluxes based on the two sensor signals were very similar (P8L335) (daytime:  $3.4 \pm 0.6 \text{ mmol m-}2 \text{ h-}1(\text{P})$ ,  $3.3 \pm 0.3 \text{ mmol m-}2 \text{ h-}1$  (Q), nighttime -0.9  $\pm$  0.1 mmol m-2 h-1 (P), -0.9  $\pm$  0.7 mmol m-2 h-1 (Q); daytime average  $3.3 \pm 0.7 \text{ mmol m-}2 \text{ h-}1$  (P, Q), nighttime average -0.9  $\pm$  0.7 mmol m-2 h-1 (P, Q)).

L233-234: I generally agree with this interpretation, it makes intuitive sense. One concern I have is that identifying what sensor works best at what time seems somewhat subjective. For instance, in Fig 6D hour 15, PAR drops from 150 to below 50 umol m-2 s-1, and sensor P registers a concurrent decrease in flux, but sensor Q does not. After hour 16 the fluxes from sensor P are clearly 'compromised', but then again, this assessment is based upon what we'd typically expect to see. I would otherwise be tempted to interpret the drop in fluxes at hour 15 in sensor P as 'real', unless there is some other metric we could use to establish flux guality. Overall, I fear that if we do not adopt some guantitative metrics for establishing flux guality beyond what we expect to see (e.g. diel dynamics in relation to PAR), then we might miss out on something new and interesting. This is especially true during the nighttime or in non-photic habitats. In the absence of light, would we be able to say with the same certainty what flux dynamics is 'true' and what isn't? We've been using a two-sensor setup since we started using eddy covariance in 2010, and I fully agree that this setup drastically increases the chances of obtaining good data. I typically evaluate the two sensor signals for their performance throughout the deployment by (a) comparing the mean O2 microsensor concentration to the O2 optode, (b) point-to-point noise in the 8 Hz data streams, and (c) linearity of the instantaneous cumulative fluxes for each 15 min flux period (Attard et al MEPS in press https://doi.org/10.3354/meps13372). Yamamoto et al (2015) L&O (https://doi.org/10.1002/lno.10018, Fig. 3) adopt a similar approach. Would fitting linear regressions to the cumulative instantaneous fluxes for each 15 min flux for sensors P and Q, and evaluating the coefficient of determination (R2 value), help to shed light on this? An additional analysis could be to fit P-I relationships to the data and see which sensor produces the best R2 value, like the approach described

**BGD**
in Attard & Glud (2020) Biogeosciences Discussions (https://doi.org/10.5194/bg-2020-140, Fig. 2).

Response: We agree with the reviewer that it would be good to develop more quantitative metrics for assessing the quality of the fluxes we measure with the EC systems. We do similar quality checks as described by the reviewer (comparison with reference sensors, noise monitoring, acoustic beam correlation checks), and we here use comparison with benthic chamber incubations and the agreement of independently measuring oxygen sensors to support the flux estimates. Although the chamber fluxes are biased due to the isolation of the incubated sediment, the magnitude of the fluxes can be considered near the true flux. A very close agreement of eddy fluxes based on two independent sensor readings further supports the fluxes we measured. Our experience is that using the linearity of the instantaneous cumulative fluxes for each 15 min flux period can be tricky and time consuming as these cumulative fluxes often change direction midway without producing artifacts and the linearity is dependent on the environment (e.g. homogeneity of the oxygen distribution in the water column) meaning that some of the observed variability is real and not a reflection of low-quality data.

L243-244: Also here, it would be good to mention what part of the integrated curve was used for this analysis.

Response: We added the information as requested by the reviewer: P9L274 "After exclusion of flux intervals compromised by biofouling (Case A: no exclusion, Case B: 14:00-18:00, Case C: 12:00-22:00, 5:00-9:00), the differences between P and Q optode fluxes derived from the slopes of the cumulative flux curves (Figs. 2E, 4E, 6E)"

Discussion

L222-271: A two sensor setup provides redundancy and cross comparison, no question about that. However, it is also twice the cost in hardware, and twice the amount of work in postprocessing. If fouling seems to be such an issue, wouldn't the right approach be to try to eliminate fouling, rather than to add more sensors? I believe there BGD
is scope in the Discussion and in the Conclusion to comment on what future modifications might be valuable. For instance, should we install a pump and back-flush the sensors before each measurement burst? Can we monitor buildup of sensor fouling in some other way?

Response: We appreciate the comment of the reviewer and added the following sentence to the conclusions: P12L357 "The deployments of the 2OEC in the Florida Keys sandflat revealed that biofouling frequently affects the aquatic eddy covariance measurements even in such an oligotrophic environment with very clear water. Further developments of the aquatic eddy covariance technique therefore may benefit from installations of devices that monitor (e.g. with a camera) and reduce or prevent biofouling (e.g. through a cleaning mechanism)."

Technical corrections L78: Remove extra 'relative'

Response: Thanks! Done.

L81: . . .established using the jet-nozzle method. . .

Response: Done.

L86: Analogue should read 'analog'

Response: Biogeosciences uses British English spelling

L127: Apostrophes should be replaced with primes

Response: Done.

L137: should read 'products of instantaneous. . .'

Response: Done.

L156: 'permitting' rather than 'facilitating'

Response: Done.
L174: Should read 'The close agreement. . .'

Response: Done.

Please also note the supplement to this comment: https://www.biogeosciencesdiscuss.net/bg-2020-172/bg-2020-172-RC2- supplement.pdf

**BGD**

---

## Author Response (AR2)

**Response to the comments and questions of the reviewers and editor List of changes made to manuscript Revised manuscript with annotations**

**Responses to the comments of associate editor Tyler Cyronak**

I'd like to see a bit more discussion on the differences between the eddy and chamber results. You offer one explanation about changing footprint, but could it also be due to natural heterogeneity of fluxes in these sediments or different advection rates under enclosed vs natural conditions?

Response: We thank Dr. Cyronak for the comments and for handling our manuscript. In response to this comment, we added the following paragraph to the discussion:

P9L449: "A perfect conformity of eddy and chamber fluxes cannot be expected due to fundamental differences between the open, non-invasive eddy covariance and the closed, invasive chamber measuring principles. Although marine sandy sediments as those investigated here may appear homogeneous, bottom current patterns and patchy colonization by e.g. algae and macrofauna cause some spatial and temporal variability in organic matter content and associated microbial metabolic activities that may influence interfacial solute fluxes (Kourelea et al., 2004;Ricart et al., 2015;Wilde and Plante, 2002;Attard et al., 2019). The fluxes recorded by the 2OEC originated from a sediment surface area of approximately 40 m2 upstream the instrument (Berg et al., 2007), and the location of this footprint area moved with flow direction. The 2OEC therefore integrated some of the natural spatial variability of the flux, and the movement of the footprint area as well as changes in bottom flow velocity are reflected in the measured fluxes. In contrast, each chamber enclosed the same surface area of about 0.03 m2 and applied the same constant pressure gradient at the sediment surface throughout each deployment. The exclusion of flow variations, temporal water composition changes, and some of the light affected the fluxes measured by the chambers. These differences between the two flux-measuring techniques may explain some of the discrepancies in dark fluxes observed between 20EC and chambers during the 16-17 Aug 2013 deployment (Fig. 8 B) and emphasize the need for including natural bottom currents and light, and integrating over larger surface areas when assessing benthic interfacial fluxes."

**Also, it is bit unclear what you are comparing in Fig. 7, is the black line the average of the 3 chamber replicates? What are the bars labeled 'Ed Ch'?**

Response: We agree that it was not clear what data we were comparing and thank the editor for pointing this out. We added the following explanation to the legend of figure 8 (former figure 7):

"Figure 8: Comparison of the day and nighttime fluxes recorded with the eddy covariance instrument and the benthic advection chambers. (a): 14-15 August deployments (Case C). (b): 16-17 August deployments (Case A). (c): Averages of the two August deployments. Light columns present daytime fluxes, dark columns nighttime fluxes. Columns on the right side of graphs (a), (b), and (c) labelled "Ed" depict the average of the fluxes based on sensors P and Q shown in the respective graph, columns labelled "Ch" depict the average of the fluxes recorded with the three chambers shown in the respective graph. Error bars represent standard deviation including error propagation. (d): Comparison of cumulative flux measured with the chambers (black line showing the average of the 3 chamber replicates and the standard deviation of the individual measuring points) and the 2OEC (red line: optode P, blue line: optode Q) during the 14-15 August deployments (Case C). The chamber fluxes confirmed that optode P was temporarily compromised during this deployment. Error bars depict standard deviation."

**Responses to the comments of reviewer Care Reimers**

Review of "Technical note: Measurements and data analysis of sediment-water oxygen flux using a new dual-optode eddy covariance instrument."

The manuscript submitted by Huettel et al. is appropriate as a technical note because it focuses on issues related to the quality of oxygen sensor measurements in the context of aquatic eddy covariance (AEC) measurements of benthic oxygen fluxes. The authors stress biases that can occur when sensors are affected by biofouling, and they illustrate with detailed examples how these artifacts can be recognized and controlled for using a dual-optode system. The examples are from an area of shallow shelf in the Florida Keys, making them unique environmentally. As a practitioner of these methods, I find this manuscript very useful, but I also recommend a number of revisions to improve clarity, especially for readers who may be less familiar with the AEC technique.

Response: We thank Dr. Reimers for the detailed review of our manuscript and the helpful comments and questions.

General recommendations: The manuscript is difficult to follow at times for reasons of organization and language. Most importantly, the introduction does not lead off with a very clear description of how biofouling or other "disturbances" can affect oxygen sensor measurements and corresponding AEC derivations. Instead the authors try to unravel these uncertain effects through the course of detailed reviews of data.

Response: We added a paragraph explaining description how disturbances including biofouling can affect measurements and corresponding AEC derivations.

P2L53: "Irrespective of the technology, the readings of the oxygen sensors can be biased by attachment of particles, bacteria or algal cells, which can affect the sensor signal through shielding of the sensor tip and metabolic processes (Smith et al., 2007;Delauney et al., 2010). Mineral particles may be impenetrable to gases, while organic particles may be sufficiently dense or oxygen consuming such that oxygen diffusion through them is reduced, thereby decreasing and delaying oxygen transport to the sensor dampens the oxygen signal and thereby reduces the calculated flux. The most common particles attaching to sensors may be marine snow particles (Fig. 1 a), sticky aggregates of various organic and inorganic particles glued together by extracellular polymeric substances (Alldredge and Silver, 1988). Bacteria and phytoplankton cells commonly contained in these particles can cause oxygen consumption and oxygen production, thereby affecting the signals of the oxygen sensor and the fluxes calculated from these readings."

We also added a sentence and figure explaining how a marine snow particle attached to the oxygen sensor can lead to increased flux estimates when waves are present.

P10L316: "A marine snow particle with photosynthesizing organisms attached to the tip of the oxygen sensor P may have caused the erroneous flux estimates. Oxygen concentration in the centre of such aggregates during light conditions can be increased by 85 % relative to the surrounding water (Ploug and Jorgensen, 1999), or even by 180% within millimetre-size gelatinous colonies of *Phaeocystis spp.*, a common global bloom-forming phytoplankton organism (Ploug et al., 1999). The movement of such an attached photosynthesizing particle by wave orbital motion can synchronize vertical current flow oscillations and the effect of the particle on the oxygen reading (e.g. increased oxygen due to photosynthesis) and thereby lead to erroneous flux estimates (Fig. 9.)"

More specific language throughout, as I will suggest below, would be helpful. Core questions are: does the biofouling produce or consume minute amounts of oxygen locally affecting what the sensor detects (sort of a contamination of the ambient condition), and why would this production or consumption be flow sensitive under waves? Zooming in to look at some data under both day and night conditions may help reveal the behavior.

Response: If one of the two parallel measuring sensors showed a temporary increase or drop in oxygen as found in the deployments on 10-11 April 2014 and 14-15 August 2013, we attributed this to the biofouling of that sensor, and in-situ inspections of the sensors revealed biofouling (extreme case now shown in Fig. 1b). We inspected the data and as an example provide the co-spectra shown in Figure 6 that reveal a temporary sensitivity to waves in sensor P, which we explain with the process now depicted in Fig. 10. Measurements have shown that marine snow particles can produce and consume substantial amounts of oxygen (see references listed in the response above) and marine snow was abundant at the study site partly due to the proximity of the coral reefs that release mucus to the water. We added this explanation to the text:

P10L311 "If one of the two parallel measuring sensors showed a temporary increase or drop in oxygen as found in the deployments on 10-11 April 2014 and 14-15 August 2013, we attributed this to the biofouling of that sensor, and in-situ inspections of the sensors revealed biofouling (extreme case now shown in Fig. 1b). Marine snow was abundant at the study site partly due to its proximity to coral reefs that release mucus to the water (Wild et al., 2004).

and

Figure 9: False flux increase caused by the rhythmical deformation of a marine snow particle attached to an oxygen fibre optode. Erroneous fluxes result when wave orbital motion modulates the distance between photosynthesising organisms contained in the gelatinous marine snow particle and the sensing surface of the optode.

It would also be helpful to simply refer to the three deployments used for illustration as something like "Case A, Case B and Case C". The dates of the deployments were so similar, that a reader has trouble differentiating the examples by date alone.

Response: We followed the suggestion of the reviewer and now use "Case A, Case B and Case C".

Specific suggestions for edits:

Figures 2, 4 and 6 panels (b) units should be micromoles per liter. (Use consistent unit designations in tables and figures).

Response: Done.

There is duplication of references: McGinnis et al. 2008a and b are the same, Reimers et al. 2012a and b are the same.

Response: Thanks for pointing this out, we removed the duplication Line by line:

Page 1 lines 9-10: First example of a vague reference to the core problem "but a main weakness of the commonly used instrumentation is the susceptibility of the delicate oxygen microsensors required for the high frequency measurements to disturbances." This needs to be rewritten. Might be best to say something like "but a critical requirement is that EC sensors are able to resolve high frequency variations in dissolved oxygen concentration and vertical velocity without artifacts." Response: We followed the suggestion of the reviewer and changed the sentence. It now reads: P1L8 In-situ fluxes can be measured non-invasively with the aquatic eddy covariance technique, but a critical requirement is that the sensors of the instrument are able to correctly capture the high frequency variations in dissolved oxygen concentration and vertical velocity".

Page 1 lines 15-17. Revise. For example as: "Short-term changes in flux were confirmed or rejected with the 2OEC, giving more certain insights into the temporal dynamics of benthic oxygen flux in permeable carbonate sands."

Response: We revised the sentence that now reads:

P1L15 "Short-term changes in flux that are dubious in measurements with single oxygen sensor instruments can be confirmed or rejected with the 2OEC and in our deployments provided new insights into the temporal dynamics of benthic oxygen flux in permeable carbonate sands."

**Page 1 line 18. Why do you say "within a couple of hours"? Do you mean that this is how much time is needed to capture a representative flux under steady conditions?**

Response: We clarified our statement following the suggestion of the reviewer. It now reads: P1L17 "Under steady conditions, representative benthic flux data can be generated with the 2OEC within a couple of hours, making this technique suitable for mapping sediment-water, intra-water column, or atmosphere-water fluxes.

Page 2 line 36. Add: Reimers et al., 2016. Microelectrode velocity effects and aquatic eddy covariance measurements under waves. J. Atm. Ocean. Tech. 33, 263-282.

Response: Done.

Page 2. lines 40-42. I question the statements: "Optodes consume no oxygen and have very low or no stirring sensitivity (Holtappels et al., 2015). Compared to micro- electrodes, they are less susceptible to signal drift and keep their calibration over longer time." It appears they may develop a stirring sensitivity once biofouled, and my experience is they may drift quite a bit due to their loss of sensitivity. Perhaps you could qualify these statements as: "Optodes consume no oxygen and may have very low or no stirring sensitivity (Holtappels et al., 2015). Compared to microelectrodes, we have observed they are less susceptible to signal drift and keep their calibration over longer time."

Response: We followed the recommendation of the reviewer, and the sentence now reads: P2L41 "Optodes consume no oxygen and may have very low or no stirring sensitivity (Holtappels et al., 2015). Compared to microelectrodes, we have observed they are less susceptible to signal drift and keep their calibration over longer time."

Page 2. lines 51-54. Here is where the authors need to give a clearer initial description of how biofouling will alter signals from an optical sensor. The statement "through shielding of the sensor tip from the water current and metabolic processes (i.e. respiration, photosynthesis)" is unclear. What kind of changes in signal magnitide and dynamics occur and why? These things are rarely "obvious", especially to new users.

Response: Thank you for pointing this out. We added the following information:

P2L53 "Irrespective of the technology, the readings of the oxygen sensors can be biased by attachment of particles, bacteria or algal cells, which can affect the sensor signal through shielding of the sensor tip and metabolic processes (Smith et al., 2007;Delauney et al., 2010). Mineral particles may be impenetrable to gases, while organic particles may be sufficiently dense or oxygen consuming such that oxygen diffusion through them is reduced, (Zetsche et al.;Ploug and Passow, 2007) thereby decreasing and delaying oxygen transport to the sensing surface. The ensuing increase in the response time of the sensor dampens the oxygen signal and thereby reduces the calculated flux. Berg et al. (2015) explained how the resulting time offset between the oxygen and the velocity data can cause significant over- or underestimation of the flux. The most common particles attaching to sensors may be marine snow particles (Fig. 1a), sticky aggregates of various organic and inorganic particles glued together by extracellular polymeric substances (EPS) (Alldredge and Silver, 1988). Bacteria and phytoplankton cells commonly contained in these particles can cause both, oxygen consumption and oxygen production, thereby affecting the signals of the oxygen sensor and the fluxes calculated from these readings. As an example, we observed oxygen flux increases up to 4.4 mmol m-2 h-1 caused by photosynthesis and decreases up to -5.2 mmol m-2 h-1 caused by respiration of microbes contained in marine snow attached to the oxygen sensor."

Page 2. line 78. Revise as "is relatively robust compared to microelectrodes"...

Response: We followed the suggestion of the reviewer, and the sentence now reads: P3L73 "With the advantages of being relatively robust compared to microelectrodes and less expensive, optodes are predisposed to become the preferable sensor-type for aquatic eddy covariance measurements"

Page 2. line 80. If the discussion of sensor drift and lifetime is based generally on previous measurements, make this clear. If it is based on the experiments in this paper, move this reporting to the results section.

Response: Sensor lifetime and drift were observed in previous field deployments. We added this information to the text:

P3L93 "Our previous field measurements indicated that when operated continuously at a measuring frequency of ~ 8 Hz, the useful lifetime of the OXR430-UHS typically was 3 to 7 days before the signal decreased to a level precluding reliable data interpretation. The signal drift over this period was negligible (< 0.03%) (Huettel, unpubl.).

Page 5. lines 136-137. Revise as: "the product of instantaneous oxygen fluctuation and instantaneous vertical velocity change" or something clearer.

Response: Done

Page 5. lines 140-146. The use of a storage term here is not well justified and later on is not clearly discussed. Is this the correction referred to in Figure 3C? Holtappels et al. (2013) illustrate transient contributions to eddy fluxes linked to changes in C, but their model predictions of these effects are different from the storage term (although both are dependent on dC/dt). At the heart of the matter is: does oxygen change due to advection or due to localized cumulative production of consumption in the bottom boundary layer? You appear to assume a changing diurnal "storage" balance in dissolved oxygen, but the oxygen time series show other drivers of change. The statement given at lines 202-204 also indicates you recognize advection.

Response: We agree with the reviewer that this was not explained sufficiently. We added the following text: P5L150 "At our measuring height of 35 cm above the seafloor, the diurnal fluctuation in mean water column oxygen concentration can result in substantial changes in the oxygen inventory of the water column below the measuring volume, which can bias the local eddy flux measurements. To correct for this effect, an oxygen storage term, calculated as  $\int_0^h dC/dt h$ , was subtracted from the measured eddy flux to determine the benthic oxygen flux (dC/dt = change of the average oxygen concentration over time, calculated through linear detrending of the measured oxygen data over 15 minute intervals, h = height of the measuring volume)(Rheuban et al., 2014b).

Page 6. line 168. Here you start referring to data processing steps as "corrections". It would help the reader if section 2.3 separated these different corrections more clearly and let the reader know their effects on flux records would be evaluated as part of the results.

Response: We agree with the reviewer and moved the effects of the flux corrections we applied to the results section.

Page 6. line 172. It is not clear what the authors mean by "over the time course of the deployment". Can they indicate over what time intervals the cumulative slope was evaluated? Did they assess the slope burst by burst, or over longer intervals? How is the standard deviation derived for these calculations?

Response: We added the following text to clarify this point:

P5L157 "For the comparison of the temporal evolution of the fluxes that were determined using the recordings of the two optodes, we calculated the cumulative fluxes over the duration of the deployments. The slopes of the increasing cumulative fluxes during daylight and decreasing cumulative fluxes during nighttime were assessed for hourly time intervals. Standard deviations of the fluxes reflect the deviations between three hourly slope determinations. All error estimates are reported as  $\pm 1$  standard deviation."

Page 7. lines 216-218. A better explanation of the signal produced by biofouling under waves needs to be given. I have seen this effect in my data too. An oscillation develops at the wave frequency that appears to be greater than what would occur if the water column gradient was moving up and down or back and forth with wave motions. Looking at segments of the oxygen, velocity and pressure time series may help sort this out. It appears to be a "velocity effect".

**Response: We added an explanation and figure 9**

P10L316 "A marine snow particle with photosynthesizing organisms attached to the tip of the oxygen sensor P may have caused the erroneous flux estimates. Oxygen concentration in the centre of such aggregates during light conditions can be increased by 85 % relative to the surrounding water (Ploug and Jorgensen, 1999), or even by 180% within the sticky millimetre-size gelatinous colonies of *Phaeocystis spp.*, a common global bloom-forming phytoplankton organism (Ploug et al., 1999). The movement of such an attached photosynthesizing particle by wave orbital motion can synchronize vertical current flow oscillations and the effect of the particle on the oxygen reading (e.g. increased oxygen due to photosynthesis) and thereby lead to erroneous flux estimates (Fig. 9)."

**Page 8. line 236. Here you discuss another reason for poor sensor performance (particle impact). This should also be mentioned in the introduction under optode weaknesses.**

Response: We added the following sentence in the introduction P2L46 "Although most optodes are more robust than microelectrodes, they can break due to particle collision." Page 9. line 286-287. State more specifically how current measurements can be affected and why. Differentiate between real changes in the flow reaching the ADV sampling volume (flow obstruction) and measurement artifacts due to acoustic returns off the sensor tip.

Response: We added the following explanation:

P11L332"A cylindrical sensor placed in the path of the flow upstream the ADV measuring volume can shed a vortex street thereby compromising the flow in the measuring volume and the flux estimates based on the flow measurements. Depending on the flow Reynolds number, such vortices may extend between 5 to 20 times the diameter of the cylinder downstream the sensor (Green, 2012). By using the Pyroscience fiber optode for the 2OEC, one of the smallest and fastest oxygen sensors presently available, potential errors caused by the disturbance of the flow and interference with the acoustic pulses of the Doppler velocimeter can be avoided. At the turbulent Reynolds numbers typical for our study site (4000 < Re < 110000), the vortices shed by the 430 µm fiber exposed to the water currents extend between 2 to 10 mm downstream of the fiber (Green, 2012). Since the tips were placed at 30 mm horizontal distance from the lower edge of the ADV measuring volume, turbulence caused by fiber-flow interaction could not reach the ADV measuring volume. Similarly, the sensor tips at that distance did not interfere with the acoustic pulses of the optode tips, we confirmed that the optode fibers did not cause any disturbances in the ADV signal."

Page 10. The paper conclusions are relatively weak. The authors could easily expand a bit on how the fluxes measured in this study compare to other inner shelf and coastal environments with permeable sediments, e.g. those of Berg et al. 2013.

Response: We agree with the reviewer that the discussion of the flux results could be expanded, however, this paper was designed to introduce the instrument and the data interpretation, and, with all due respect, decided not to expand the discussion of the flux results in this paper. We are presently working on a manuscript that uses the results from these deployments together with other flux data measured at this study site to demonstrate the high metabolic activity of the coarse carbonate sands and to discuss their role in the coral reef ecosystem. This paper will also include a comparison of the fluxes presented here with fluxes measured in other inner shelf environments. We expanded the Conclusion sections with material directly related to the method development presented in this paper:

P12L359"We propose using the agreement/disagreement between the fluxes calculated from the signals of two independently measuring optodes as a tool to assess the quality of the measured fluxes. The nearly identical cumulative fluxes calculated from the two optodes in our August (Case A) and April (Case B) deployments strongly imply that the dynamics of the fluxes were measured accurately by the system. Likewise, the near linearity of the cumulative flux increase during daytime and decrease during nighttime (Figs. 3e, 5e) and the very similar slopes of these cumulative flux curves support that the measurements recorded representative fluxes. The good agreement of the fluxes measured with the eddy covariance instrument and the fluxes measured independently with a very different method (advection chambers, Fig. 8a, b, d) indicate that the magnitudes of the fluxes recorded by the 2OEC were correct. The deployments of the 2OEC in the Florida Keys sandflat revealed that biofouling frequently affects the aquatic eddy covariance measurements even in such an oligotrophic environment with very clear water containing low amounts of phytoplankton, bacteria and particles. Further developments of the aquatic eddy covariance technique therefore may benefit from installations of devices that monitor (e.g. with a camera) and reduce or prevent biofouling (e.g. through a cleaning mechanism). This project intended improving the reliability of the aquatic eddy covariance technique and the procedures of data analysis in order to promote this powerful technique."

**Responses to the comments of reviewer Karl Attard**

**General comments**

Huettel et al. present a technical study describing a new dual-optode eddy covariance system. The authors integrate two independent O2 sensors within a standard eddy covariance setup to cross-check fluxes extracted using two independent O2 sensor output streams, and to identify any biases in the measurements which are most likely caused by sensor fouling. Dual O2 sensor eddy systems are not new per se (e.g. McGinnis et al. 2011, Attard et al. 2014), but it is the first time that the two sensor signals have been compared in the level of detail provided in this study. The authors also perform chamber incubator measurements in parallel with eddy covariance to resolve O2 fluxes using two different state-of-the-art methods. Finally, the authors also provide a comparison between the three most popular O2 sensor systems for eddy covariance measurements.

The paper by Huettel et al will find broad interest among the growing community of aquatic eddy covariance users. The length of the paper and the angle of the study make it appropriate to be published as a Technical Note in Biogeosciences. The scientific methods are clearly outlined, language is fluent and precise, referencing is appropriate and up-to-date, and the overall presentation is well-structured and clear. I have one main comment and several smaller comments that the authors may wish to address.

Response: We thank Dr. Attard for the detailed review of our manuscript and the helpful comments and questions.

My main comment concerns how flux quality is evaluated. Currently, the authors determine quality based on (a) diel dynamics of O2 fluxes in relation to PAR, and (b) by comparison to chamber incubator measurements. If the quality-checking aspects could be expanded to include other metrics, then I foresee that the dual sensor approach would be useful in a broader range of settings.

Response: We agree with the reviewer that assessing the quality, validity and accuracy of the fluxes is central when conducting non-invasive eddy covariance measurements. In addition to the two methods mentioned by the reviewer (i.e. diel flux dynamics in relation to PAR, and comparison to chamber measurements), we use here the agreement/disagreement between the fluxes calculated from the signals of two independently measuring optodes as a tool to assess the quality of the measured fluxes. To make that point more clear, we added the following text to the conclusions:

P12L359 "We propose using the agreement/disagreement between the fluxes calculated from the signals of two independently measuring optodes as a tool to assess the quality of the measured fluxes. The nearly identical cumulative fluxes calculated from the two optodes in our August (Case A) and April (Case B) deployments strongly imply that the dynamics of the fluxes were measured accurately by the system. Likewise, the linearity of the cumulative flux increase during daytime and decrease during nighttime (Figs. 3e, 5e) and the very similar slopes of these cumulative flux curves support that the measurements recorded representative fluxes. The good agreement of the fluxes measured with the eddy covariance instrument and the fluxes measured with independently with a very different method (advection chambers, Fig. 8a, b, d) indicate that the magnitudes of the fluxes recorded by the 20EC are correct."

**Specific comments Introduction**

L20-67: It would be fair to mention that dual O2 sensor eddy systems have been in use for years (e.g. McGinnis et al. 2011 L&O Methods, Attard et al. 2014 L&O) but that so far, no detailed comparison between sensor signal output has been presented.

Response: We thank the reviewer for pointing out the missing references to other dual sensor instruments. In a earlier version of the manuscript, we had an extended discussion of microlectrode-equipped eddy instruments, which included the dual electrode systems deployed by Attard et al (2014) and McGinnis et al. (2011). This matter was removed when shortening the paper and we now re-inserted these references. To the best of our knowledge, eddy instruments based on dual optode measurements have not been introduced, and we are not aware of publications that use the date evaluation approach explained here. The following section was inserted into the text: P2L35 "To improve the reliability of the flux measurements, eddy covariance with dual microelectrodes were developed, e.g. (Attard et al., 2014;McGinnis et al., 2011;Rodil et al., 2019;de Froe et al., 2019;Rovelli et al., 2015)"

**Methods**

L70: It is worth mentioning that these meters only use half the potential voltage range of the Vector analog channels (0-2.5V), but developments are in place to increase this to the full range (0-5V).

Response: We thank the reviewer for this suggestion and added the following sentence to Table A1 in Appendix A: "the meter presently uses half the potential voltage range of the Vector analog input, but developments are in place to increase this to the full range (0-5V)"

**L71: I cannot find this model on the Pyroscience website. Do you mean the FSO2- SUBPORT? https://www.pyroscience.com/en/products/all-meters/fso2-subport**

Response: We thank the reviewer for pointing this out. The Pyroscience™ FireStingO2-Mini oxygen meter recently has been replaced by the Pyroscience™ PICO-O2, which is similar to the meter we used but more compact. The FireStingO2-Mini oxygen meter is still available in combination with the Pyroscience™ FSO2-SUBPORT. We added this information to the text:

P3L84The eddy covariance instrument we developed uses two Pyroscience™ FireSting O2-Mini oxygen meters (specifications listed in Table A1 in Appendix A, now sold in combination with the Pyroscience™ subport (FSO2-SUBPORT)) that read two ultra-high speed Pyroscience™ OXR430-UHS retractable oxygen minisensors (Table A2).

We also added the following text to the legend of Table A1 in Appendix A:

"This oxygen meter recently has been replaced by the Pyroscience™ PICO-O2, which is similar to the FireStingO2-Mini but more compact. The FireStingO2-Mini oxygen meter is still available in combination with the Pyroscience™ subport (FSO2-SUBPORT)."

L86-87: It would be useful to specify whether you powered the analog channels through the Vector

Response: Thank you for this suggestion. We added P3L86 "The FireSting O2-Mini oxygen meters were supplied with the output power of the ADV (see below)." L95: Firesting O2Mini: Again here, please check that this is the right model, or specify whether this was an older model that has since been replaced by the FSO2-SUBPORT.

Response: To avoid confusion we added the following text: P3L84 "(specifications listed in Table A1 in Appendix A, now sold in combination with the Pyroscience™ subport (FSO2-SUBPORT)"

L97: 30 mm is quite a large distance. Any reason for not moving closer to the measurement volume (e.g. 1 cm)?

Response: We chose this distance to prevent any potential interference with the flow and acoustic pulses of the Vector ADV. We added the following sentence to make this clear P4L114 "This distance prevents any disturbance of the flow in the measuring volume of the ADV and any interference with the acoustic pulses of the Vector."

L143-145: This reads like Results.

Response: We agree and moved this sentence to the results section.

L148-161: It would be useful to describe the chamber measurements in more detail. How were the chamber O2 fluxes calculated? Did you have an optode inside the chamber measuring O2 concentration continuously? How long did the deployments last? Ultimately, what are we comparing in Fig. 7?

Response: We added the following paragraph providing more detail about the chamber measurements: P6L169 "For the deployments at our study site, the pressure gradient in the chambers was set to 0.2 Pa cm-1, corresponding to the gradient produced by a 10 cm s1 bottom flow deflected by a ripple of 7 cm height (Huettel and Gust, 1992a). In highly permeable sediments, the pressure gradient in the chamber causes pore water flow through the surface layer of the enclosed sediment, thereby mimicking the pore water exchange occurring in the surrounding rippled seabed. The transparent chamber and stirring disk allow penetration of light to the enclosed sediment (~10% loss in PAR through light attenuation caused by the acrylic), permitting benthic photosynthesis in the chamber. The acrylic cylinder of the chambers was pushed 12 cm into the sand sediment, resulting in a chamber water volume of 5 L. A Hach Rigid O2 Optode mounted in the chamber lid collected oxygen concentrations at 15 minute time intervals. The fluxes were calculated from the changes of the oxygen concentration in the water column of the chamber over time. Chamber incubations ran for 24 h, then the lid was opened to allow re-equilibration with the ambient water before starting the next measurement cycle."

We added to the legend of figure 8 (former figure 7) "Comparison of cumulative flux measured with the chambers (black line showing the average of the 3 chamber replicates and the standard deviation of the individual measuring points) and the 2OEC (red line: optode P, blue line: optode Q) during the 14-15 August deployments (Case C). The chamber fluxes confirmed that optode P was temporarily compromised during this deployment. Error bars depict standard deviation."

L156: Do the chambers attenuate PAR? Results

Response: Thank you for asking this question. We measured a 10% loss in PAR caused by the acrylic chamber lids and added this information to the text

P6L174 "(~10% loss in PAR through light attenuation caused by the acrylic)"

**L164: The first sentence seems out of place here.**

Response: We structured the Results sections by these subtitles, and the purpose of these subtitles becomes more clear when reading the related paragraphs.

**L171-172: What deployment hours did you use for this analysis? Was it all of daylight hours i.e. until approx. 20:00? If so, does linear regression adequately represent these dynamics?**

Response: We used all daylight hours and the flux increase during daylight could be represented well by linear regression ( $R^2 > 0.9$ ). The same applied to the nighttime fluxes. When calculating the fluxes, we avoided sections with disturbances in the cumulative flux curves. We added the following explanation to the text: P7L194 "The flux increase during daylight and decrease during nighttime could be represented well by linear regression ( $R^2 > 0.9$ )."

**L179-183: I would expect that both optodes located at 35 cm above the seafloor and 1 cm apart would capture these variations, though?**

Response: They do capture these variations, however, the higher heterogeneity of the oxygen distribution in the water still causes larger differences between the simultaneous readings of the two optodes during daylight hours (Fig. 4a)

L231: Fig 6E: I suppose that the jump at hour 22 in the cumulative flux for sensor P is not real, but it was offset in post processing to indicate that the two sensors match one another very well beyond this point. I understand the wish to illustrate this, but I think it is confusing, because it suggests that despite the fluxes from both sensors being very different prior to hour 22, the daily integrated flux is very similar, which cannot be the case.

Response: With all due respect, we disagree with the reviewer here. As pointed out in the legend of the figure, P cumulative flux at 18:00 was intentionally reduced by 5 mmol m-2 to allow comparison of the two cumulative fluxes based on P and Q data (Panel E). After excluding the time period during which sensor P was compromised by biofouling, the daily integrated fluxes based on the two sensor signals were very similar (P8L335) (daytime:  $3.4 \pm 0.6 \text{ mmol m}^{-2} \text{ h}^{-1}(\text{P})$ ,  $3.3 \pm 0.3 \text{ mmol m}^{-2} \text{ h}^{-1}(\text{Q})$ , nighttime  $-0.9 \pm 0.1 \text{ mmol m}^{-2} \text{ h}^{-1}(\text{P})$ ,  $-0.9 \pm 0.7 \text{ mmol m}^{-2} \text{ h}^{-1}(\text{Q})$ , nighttime average  $-0.9 \pm 0.7 \text{ mmol m}^{-2} \text{ h}^{-1}(\text{P}, \text{Q})$ ).

L233-234: I generally agree with this interpretation, it makes intuitive sense. One concern I have is that identifying what sensor works best at what time seems somewhat subjective. For instance, in Fig 6D hour 15, PAR drops from 150 to below 50 umol m- 2 s-1, and sensor P registers a concurrent decrease in flux, but sensor Q does not. After hour 16 the fluxes from sensor P are clearly 'compromised', but then again, this assessment is based upon what we'd typically expect to see. I would otherwise be tempted to interpret the drop in fluxes at hour 15 in sensor P as 'real', unless there is some other metric we could use to establish flux quality. Overall, I fear that if we do not adopt some quantitative metrics for establishing flux quality beyond what we expect to see (e.g. diel dynamics in relation to PAR), then we might miss out on something new and interesting. This is especially true during the nighttime or in non-photic habitats. In the absence of light, would we be able to say with the same certainty what flux dynamics is 'true' and what isn't?

We've been using a two-sensor setup since we started using eddy covariance in 2010, and I fully agree that this setup drastically increases the chances of obtaining good data. I typically

evaluate the two sensor signals for their performance throughout the deployment by (a) comparing the mean O2 microsensor concentration to the O2 optode, (b) point-to-point noise in the 8 Hz data streams, and (c) linearity of the instantaneous cumulative fluxes for each 15 min flux period (Attard et al MEPS in press https://doi.org/10.3354/meps13372). Yamamoto et al (2015) L&O (https://doi.org/10.1002/lno.10018, Fig. 3) adopt a similar approach. Would fitting linear regressions to the cumulative instantaneous fluxes for each 15 min flux for sensors P and Q, and evaluating the coefficient of determination (R2 value), help to shed light on this? An additional analysis could be to fit P-I relationships to the data and see which sensor produces the best R2 value, like the approach described in Attard & Glud (2020) Biogeosciences Discussions (https://doi.org/10.5194/bg-2020-140, Fig. 2).

Response: We agree with the reviewer that it would be good to develop more quantitative metrics for assessing the quality of the fluxes we measure with the EC systems. We do similar quality checks as described by the reviewer (comparison with reference sensors, noise monitoring, acoustic beam correlation checks), and we here use comparison with benthic chamber incubations and the agreement of independently measuring oxygen sensors to support the flux estimates. Although the chamber fluxes are biased due to the isolation of the incubated sediment, the magnitude of the fluxes can be considered near the true flux. A very close agreement of eddy fluxes based on two independent sensor readings further supports the fluxes we measured. Our experience is that using the linearity of the instantaneous cumulative fluxes for each 15 min flux period can be tricky and time consuming as these cumulative fluxes often change direction midway without producing artifacts and the linearity is dependent on the environment (e.g. homogeneity of the oxygen distribution in the water column) meaning that some of the observed variability is real and not a reflection of low-quality data.

**L243-244: Also here, it would be good to mention what part of the integrated curve was used for this analysis.**

Response: We added the information as requested by the reviewer:

P9L269 "After exclusion of flux intervals compromised by biofouling (Case A: no exclusion, Case B: 14:00-18:00, Case C: 12:00-22:00, 5:00-9:00), the differences between P and Q optode fluxes derived from the slopes of the cumulative flux curves (Figs. 2E, 4E, 6E)"

**Discussion**

L222-271: A two sensor setup provides redundancy and cross comparison, no question about that. However, it is also twice the cost in hardware, and twice the amount of work in postprocessing. If fouling seems to be such an issue, wouldn't the right approach be to try to eliminate fouling, rather than to add more sensors? I believe there is scope in the Discussion and in the Conclusion to comment on what future modifications might be valuable. For instance, should we install a pump and back-flush the sensors before each measurement burst? Can we monitor buildup of sensor fouling in some other way?

Response: We appreciate the comment of the reviewer and added the following sentence to the conclusions: P12L366 "The deployments of the 2OEC in the Florida Keys sandflat revealed that biofouling frequently affects the aquatic eddy covariance measurements even in such an oligotrophic environment with very clear water. Further developments of the aquatic eddy covariance technique therefore may benefit from installations of devices that monitor (e.g. with a camera) and reduce or prevent biofouling (e.g. through a cleaning mechanism)."

Technical corrections L78: Remove extra 'relative'

Response: Thanks! Done.

L81: . . . established using the jet-nozzle method. . .

Response: Done.

L86: Analogue should read 'analog'

Response: Biogeosciences uses British English spelling

L127: Apostrophes should be replaced with primes

Response: Done.

L137: should read 'products of instantaneous. . .'

Response: Done.

L156: 'permitting' rather than 'facilitating'

Response: Done.

L174: Should read 'The close agreement. . .'

Response: Done.

End of the response to the comments and questions of the reviewers and editor

**List of changes made to the manuscript**

- Added a more detailed discussion on the differences between the eddy and chamber results.
- Expanded legend of Figure 8 to clarify which data are shown in the figure.
- Added a description how disturbances including biofouling can affect measurements and corresponding AEC derivations. We included a figure explaining how a marine snow particle attached to the oxygen sensor can lead to increased flux estimates when waves are present.
- We labeled the deployments "Case A, Case B and Case C" to improve readabilility.
- We clarified our statement regarding optode stirring sensitivity
- A statement was added that advection is recognized as a factor that can influence the flux measurements.
- We now explain in more detail how current measurements can be affected by the presence of the oxygen sensor(s)
- Added a statement how the agreement/disagreement between the fluxes calculated from the signals of two independently measuring optodes can be used as a tool to assess the quality of the measured fluxes.
- We included references to previous investigations that used eddy systems with two electrodes.
- We added information regarding the new model of the pyroscience oxygen meter.
- Added a justification for the distance between oxygen sensor and ADV measuring volume.
- Added information on light attenuation by the chamber lid.
- We added an explanation on how the dual sensor approach can improve data reliability.

In addition to these major changes listed above, numerous small changes (word replacements, typo corrections, word placements) were made that are listed in the attached word documents showing the changes made during the revision.

**Technical note: Measurements and data analysis of sediment-water oxygen flux using a new dual-optode eddy covariance instrument.** Markus Huettel1, Peter Berg2, Alireza Merikhi1**

1 Department of Earth, Ocean and Atmospheric Science, Florida State University, Tallahassee, FL 32306-4520, USA

2 Department of Environmental Sciences, University of Virginia, Charlottesville, VA 22904-4123, USA

Correspondence to: Markus Huettel (mhuettel@fsu.edu)

Abstract. Sediment-water oxygen fluxes are widely used as a proxy for organic carbon production and mineralization at the seafloor. In-situ fluxes can be measured non-invasively with the aquatic eddy covariance technique, but a critical requirement is that the sensors, of the instrument are able to correctly capture the high frequency variations in dissolved oxygen

- 10 concentration and vertical velocity, Even small changes in sensor characteristics during deployment as caused e.g. by biofouling can result in erroneous flux data. Here we present a dual-optode eddy covariance instrument (2OEC) with two fast oxygen fibre, sensors and document how erroneous flux interpretations and data loss can effectively be reduced by this hardware and a new data analysis approach. With deployments over a carbonate sandy sediment in the Florida Keys and comparison with parallel benthic advection-chamber incubations, we demonstrate the improved data quality and data reliability
- 15 facilitated by the instrument and associated data processing. Short-term changes in flux that are dubious in measurements with single oxygen sensor instruments can be confirmed or rejected with the 2OEC and in our deployments provided new insights into the temporal dynamics of benthic oxygen flux in permeable carbonate sands. Under steady conditions, representative, benthic flux data can be generated with the 2OEC within a couple of hours, making this technique suitable for mapping sediment-water, intra-water column, or atmosphere-water fluxes.

**Deleted: commonly used instrumentation is the susceptibility of**

the delicate oxygen microsensors required for

[revised manuscript text omitted]

this period was negligible (< 0.03%). The optodes for the in-situ measurements were selected for similar fast response times (< 300 ms) established using the jet-nozzle method introduced by Merikhi et al. (2018). The acoustic Doppler velocimeter

145 (ADV) used for this eddy instrument was a NORTEK Vector, which is a single-point current meter capable of measuring velocity and current direction in a small measuring volume (14 mm diameter, 14 mm height (user-specified)), at rates up to 64 Hz (Table A3). Together with the current flow measurements, the Vector records pressure and temperature, as well as the compass direction and tilt of the instrument. The internal data logger of the Vector stored the current velocity data simultaneously with the two analogue signals produced by the O2Mini oxygen meters. An external battery with a capacity of 200 Wh connected to the Vector provided power for continuous measurements of up to one week duration.

The instruments were mounted on a tripod (width 120 cm, height 100 cm, Fig. 2), made of rectangular 304-stainless steel tubes (2 cm x 2 cm cross-section), with legs consisting of stainless steel rods, 1.3 cm in diameter with 20 cm diameter base plates. An extension arm held the ADV in the centre of the frame. The underwater housing (AGO Environmental Electronics) 155 containing the oxygen meters with supply voltage regulator (Dimension Engineering) and the external battery pack (4 x NORTEK Lithium-Ion 12 V, 50 Wh) were attached to the horizontal upper bar of the tripod. All electrical cables used ImpulseTM wet pluggable micro inline connectors. The two optodes were linked through two custom-made (Huettel)

underwater housing fibre-feed-through plugs with standard ST-connectors to the FireSting O2Minis. A stainless steel rod (8 mm diam.) with adjustable holders and aligned with the X-direction of the ADV, positioned the two optodes parallel to each
other and at a 45-degree downward angle. The sensing tips of the optodes were 10 mm apart from each other and located at 30 mm horizontal distance from the lower edge of the ADV measuring volume (Fig. 2b). This distance prevents any disturbance of the flow in the measuring volume of the ADV and any interference with the acoustic pulses of the Vector, A PAR-light sensor (Odyssey® Submersible Photosynthetic Active Radiation Logger) installed above the ADV logged light intensity at 5 minute intervals throughout the deployments. An Aanderaa Seaguard RCM multisensor, probe, installed with its sensors at the same height as the ADV measuring volume at 5 m distance from the tripod, recorded oxygen and temperature reference data.

**2.2 Field tests**

The performance of the 2OEC was tested through three deployments on 14-15 August 2013 (Case C), 16-17 August 2013 (Case A), and 10-11 April, 2014 (Case B) in a subtropical inner shelf environment with relatively constant salinity (35-36) and temperatures (April: 25° ± 0.8°C, August: 30° ± 0.5°C) approximately 9 km south of Long Key in the Florida Keys (24° 43.52'N, 80° 49.85'W). The site was located at 9 ± 1 m water depth near the centre of a large flat carbonate platform covered with coral sand. The unobstructed, fairly steady current flows across the platform and the relatively uniform surface roughness (ripple topography < 10 cm) produced similar turbulent diffusivity throughout the deployments. The highly permeable carbonate sand (permeability: k = 3×10-11 ± 0. 2×10-11 m2) had a median grain size of 440 µm and was inhabited by microphytobenthos (2-6 µg Chl. a g-1 sed. dw) and sparsely distributed (< 20 m-2) *Halimeda* sp. Macroalgae (Fig. 2A). In the

175 clear water (Turbidity < 8 NTU) light intensities at the seafloor reached up to 300  $\mu$ E m-2 s-1. The current flow conditions

| Deleted: | center                                        |
|----------|-----------------------------------------------|
|          |                                               |
|          |                                               |
|          |                                               |
| Deleted: | fiber                                         |
|          |                                               |
| Deleted: | such that their measuring                     |
| Deleted: | were                                          |
| Deleted: | and 10 mm horizontally apart from each other. |
|          |                                               |
| Deleted  | multiconcom                                   |

| (      | Deleted: (         |
|--------|--------------------|
| (      | Deleted: 14-15 and |
| (      | Deleted: ,         |
| $\geq$ | Deleted: April     |
| (      | Deleted: center    |

190

during all deployments were moderate (average mean flow velocity 5 to 20 cm s-1, significant wave height < 0.7 m), and the weather was generally sunny with some scattered clouds. Prior to the deployments, the oxygen optodes were calibrated in ambient seawater (water bubbled with air or with sodium sulphite addition), with the calibration data stored on the Vector logger. The measuring volume of the ADV was adjusted to be ~35 cm above the sediment-water interface. SCUBA divers positioned the instrument at the seafloor such that the Vector's X-direction was aligned with the main bottom flow direction, which was in northeast-southwest direction. The instrument was typically deployed in the morning at 9:00-10:00 and retrieved 195 24 h later. During the first hour after deployment, no flux data were collected to allow temperature adjustment of the instruments. Before downloading the data from the Vector, the calibration of the oxygen sensors was repeated and stored with

**2.3 Data processing**

the data file.

- Velocity data with acoustic beam correlations < 50% were replaced through linear interpolation of the neighbouring velocity 200 values. Oxygen data were not cleaned or despiked prior to flux calculations. Oxygen fluxes were calculated using EddyFlux 3.2 software package (P. Berg) as follows: Vertical velocity data and oxygen concentration data were reduced from 64 Hz to 8 Hz through averaging, which lessened data noise while maintaining sufficient resolution for resolving high-frequency eddies. The fluctuating component of the oxygen concentrations was determined through Reynolds decomposition, i.e. oxygen base concentrations were determined for 15 min intervals through linear detrending and subtracted from the instantaneous oxygen
- 205 data to arrive at the instantaneous oxygen fluctuations  $O_2'$  Instantaneous vertical velocity changes  $V'_{v}$  were determined through Reynolds decomposition analogous to the oxygen fluctuations. The time lag caused by the 30 mm horizontal distance between flow and oxygen measurement locations were corrected according to Berg et al. (2015) through applying time shift corrections that yielded most negative (night) or most positive (day) cross-correlations of the oxygen fluctuation and vertical movement. Oxygen fluxes then were calculated by averaging over time the product of instantaneous oxygen fluctuation and
- 210 instantaneous vertical velocity change:  $O_2 Flux = \overline{O_2 \times V_2}$  (Berg et al., 2003). At our measuring height of 35 cm above the seafloor, the diurnal fluctuation in mean water column oxygen concentration can result in substantial changes in the oxygen inventory of the water column below the measuring volume, which can bias the local eddy flux measurements. To correct for this effect, an oxygen storage term, calculated as  $\int_0^h dC/dt$  h, was subtracted from the measured eddy flux to determine the benthic oxygen flux (dC/dt = change of the average oxygen concentration over time, calculated through linear detrending of
- 215 the measured oxygen data over 15 minute intervals, h = height of the measuring volume)(Rheuban et al., 2014b), Acceleration or deceleration of current flows can alter the oxygen concentration profile and thereby modulate vertical flux (Holtappels et al., 2013). Our measurements indicated that the temporal flux variations caused by transient velocity changes Jargely cancelled. out over time (Rheuban et al., 2014b), and A correction for transient velocity changes was not applied. For the comparison of the temporal evolution of the fluxes that were determined using the recordings of the two optodes, we calculated the cumulative 220

**fluxes over the duration of the deployments. The slopes of the increasing cumulative fluxes during daylight and decreasing**

**Deleted: O2'**

**Deleted: change Vz'**

Deleted: As a tilted position of the eddy instrument relative to the main flow direction can bias the magnitude of the vertical flux, we tested whether fluxes were influenced by tilt by rotating the average flow velocity vectors for 15 min intervals until the average Vz equalled zero. These tests indicated that tilt corrections were not required for our three deployments. Similarly, we tested whether wave rotation influenced flux by rotating flow velocity vectors until SD(vy) and SD(vz)(SD=1 Standard Deviation) reached a minimum. Correction for wave rotation had a small effect and was applied in all data sets Deleted: and displacement of the vertical oxygen gradient by wave orbital motion Deleted: (2015) Deleted: products Deleted: Because large-scale variations in the average Deleted: (e.g. as caused by the diurnal changing balance of production and consumption processes) may not be captured by the turbulent. Deleted: (Rheuban et al., 2014b), Deleted: added to the eddy Deleted: ) Deleted: These Deleted: may

**Deleted: cancel**

cumulative fluxes during nighttime were assessed for hourly time intervals. Standard deviations of the fluxes reflect the deviations between three hourly slope determinations. All Error estimates are reported as ±1 standard deviation.

**2.4 Advection chamber deployments**

- In August 2013, 3 advection chambers were deployed parallel to the eddy covariance instrument to allow comparison with an independent flux data set produced by a different method. Benthic advection chambers present an in-situ incubation technique that can account for some of the current and light effects influencing benthic flux (Janssen et al., 2005a;Huettel and Gust, 1992b), The rotation of a stirring disk (15 cm diam.) within these cylindrical chambers (30, cm height, 19 cm inner diameter) produces, a radial pressure gradient at the surface of the enclosed sediment that is similar in magnitude to the pressure gradient
  generated by bottom currents interacting with present ripple topography (Huettel and Rusch, 2000). For the deployments at our study site, the pressure gradient in the chambers was set to 0.2 Pa cm-1, corresponding to the gradient produced by a 10 cm
- s-1 bottom flow deflected by a ripple of 7 cm height (Huettel and Gust, 1992a). In highly permeable sediments, the pressure gradient in the chamber causes pore water flow through the surface layer of the enclosed sediment, thereby mimicking the pore water exchange occurring in the surrounding rippled seabed. The transparent chamber and stirring disk allow penetration of
- 265 light to the enclosed sediment (~10% loss in PAR through light attenuation caused by the acrylic), permitting benthic photosynthesis in the chamber. The acrylic cylinder of the chambers was pushed 12 cm into the sand sediment, resulting in a chamber water volume of 5 L. A Hach Rigid O2 Optode mounted in the chamber lid collected oxygen concentrations at 15 minute time intervals. The fluxes were calculated from the changes of the oxygen concentration in the water column of the chamber over time. Chamber incubations ran for 24 h, then the lid was opened to allow re-equilibration with the ambient water
- 270 before starting the next measurement cycle. Although flow, light and water composition changes within the chamber are not identical to the external conditions and cause an inherent bias, the daily fluxes measured by these chambers are considered to be close (within a factor ~2) to the true fluxes (DeBeer et al., 2005;Cook et al., 2007;Janssen et al., 2005a), and this technique has been deployed successfully in numerous investigations of shallow permeable sediments (Huettel and Gust, 1992b;Eyre et al., 2013;Eyre et al., 2013;Eyre et al., 2013;Santos et al., 2011;Janssen et al., 2005b).

**275 3 Results**

280

**3.1 Instrument deployments**

The 2OEC improves the reliability of measured fluxes. The deployment of 16-17 August 2013 (Case A, Fig. 3) was characterized by moderate bottom currents averaging  $3.6 \pm 2.2$  cm s-1 (35 cm above sediment) with sustained peak velocities of 8.0 cm s-1 (Fig. 3A) and relatively low light intensities at the seafloor < 100  $\mu$ E m-2 s-1 during daytime hours (Fig. 3E). The good agreement of the independent O2 readings of both fibre optodes and the Seaguard reference optode (Fig. 3B) implied that the optodes maintained their calibration throughout the deployment. Identical corrections were applied to P and Q optode data sets when calculating fluxes (Fig. 3C), which included corrections for change in average water oxygen concentration, time lag,

| ( | Deleted: (Janssen et al., 2005a; Huettel and Gust, 1992).                        |
|---|----------------------------------------------------------------------------------|
| ( | Deleted: 20                                                                      |
|   | Deleted: diam.) is set according to flow and ripple dimensions to produce |
| l | Deleted: (0-1 Pa cm -1 )                                       |
| ( | Deleted: that produced                                                           |

| •{ | Deleted: |   | facilitatin |
|----|----------|---|-------------|
| х  | Deleteur | • | racintatin  |

| _(     | Deleted: 2A    |
|--------|----------------|
| (      | Deleted: 2E    |
| (      | Deleted: fiber |
| $\geq$ | Deleted: 2B    |
| (      | Deleted: 2C    |

and wave rotation. The conformity of the 15 min cumulative fluxes calculated from the two fibre optode signals (Fig. 3D) and the agreement of the cumulative flux curves over the time course of the deployment (Fig. 3E) corroborated the flux estimates. The flux increase during daylight and decrease during nighttime could be represented well by linear regression (R2 > 0.9). The slopes of the cumulative flux curves over the time course of the deployment (Fig. 3E) revealed daytine fluxes of 3.4 ± 0.6 mmol m-2 h-1 (P) and 3.4 ± 0.4 mmol m-2 h-1 (Q) and nighttime fluxes of -1.3 ± 0.9 mmol m-2 h-1 (P) and -1.6 ± 0.9 mmol m2.
h-1 (Q). The close agreement between the fluxes calculated from the two optode signals supported an average daytine flux of 3.4 ± 0.7 mmol m-2 h-1 (P, Q) and nighttime fluxes of -1.4 ± 1.3 mmol m-2 h-1 (P, Q) for Case Ay

Analysis of the differences between optode P and optode Q based fluxes indicated that changes in environmental settings as well as changes in optode characteristics produced the discrepancies. Larger differences between the P and Q 15 minute fluxes

- 310 were observed during daytime and when fluxes were near zero and changing direction at sunset (Fig. 4A). Patchy distribution of microphytobenthos and its photosynthetic oxygen production may result in a more uneven oxygen distribution in the bottom currents during daytime (Bartoli et al., 2003;Jesus et al., 2005;MacIntyre et al., 1996). Likewise, the patchy distribution of macrofauna and its activity peak near sunset (Wenzhofer and Glud, 2004) may be responsible for enhanced heterogeneity in the oxygen distribution in the bottom currents and ensuing larger differences between the parallel-measured fluxes at sunset.
- 315 Figure 4BrF depict the effects of the corrections that were equally applied to optode P and Q data sets to account for changes in environmental parameters and time lag error when calculating the respective fluxes. Corrections for instrument tilt were not required for our three deployments, and rotating the average flow velocity vectors did not produce significant changes in the fluxes. During the first 4 hours of the deployment, the raw, unprocessed cumulative fluxes (no corrections applied) derived from both optode signals were nearly identical before differences increased (Fig. 4B). Correction for temporal change in the
- 320 average water oxygen concentration (Fig. 4C), led to slight rate increases in the cumulative fluxes during the day as well as during the night. The correction for time lag between current flow and oxygen signal had an effect mostly during the last 7 h of the deployment (Fig. 4D), possibly due to a minor growth of biofilm on the optodes. Correction for wave rotation caused a small rate increase in fluxes, which was more pronounced during nighttime (Fig. 4E). Simultaneous application of the above corrections resulted in a nearly perfect agreement between the cumulative fluxes calculated from the two optode signals (Fig.

325 4F). Temporal flux variations caused by transient velocity changes did not have a significant impact on the cumulative flux as indicated by the good agreement between the fluxes derived from the eddy covariance measurements and those recorded in parallel benthic advection chamber measurements as reported below,

The parallel optode measurements confirmed short-term changes, e.g. the concentration step in the oxygen record at 18:14,
caused by the change of the tide (high tide: 18:16) and associated change in flow direction. The slower Seaguard oxygen sensor did not pick up this abrupt step. The temporarily increased benthic oxygen consumption near 20:00, coinciding with sunset, may have resulted from decomposition of highly degradable photosynthesis products accumulating in the sediment during daytime (Koopmans et al., 2020) and the aforementioned activity burst of the macrofauna at sunset (Wenzhofer and Glud,

[revised manuscript text omitted]

| Deleted: ,           Deleted: the August 16-17, 2013 deployment,           Deleted: 4A           Deleted: fiber           Deleted: fiber           Deleted: 4B           Deleted: 4C           Deleted: 5           Deleted: 4E           Deleted: 4E           Deleted: (P):           Deleted: (Fig. 6           Deleted: 6B | Deleted: ,         Deleted: the August 16-17, 2013 deployment,         Deleted: 4A         Deleted: fiber         Deleted: fiber         Deleted: 4B         Deleted: 4C         Deleted: 5         Deleted: 5         Deleted: 4E         Deleted: (P):         Deleted: ,         Deleted: 6B         Deleted: 6E |          |                                    |
|--------------------------------------------------------------------------------------------------------------------------------------------------------------------------------------------------------------------------------------------------------------------------------------------------------------------------------|---------------------------------------------------------------------------------------------------------------------------------------------------------------------------------------------------------------------------------------------------------------------------------------------------------------------|----------|------------------------------------|
| Deleted: the August 16-17, 2013 deployment,           Deleted: 4A           Deleted: fiber           Deleted: fiber           Deleted: 4B           Deleted: 5           Deleted: 5           Deleted: 4E           Deleted: (P):           Deleted: (Fig. 6           Deleted: 6B                                             | Deleted: the August 16-17, 2013 deployment,           Deleted: 4A           Deleted: fiber           Deleted: fiber           Deleted: 4B           Deleted: 4C           Deleted: 5           Deleted: 5           Deleted: 4E           Deleted: (P):           Deleted: (Fig. 6           Deleted: 6B            | Deleted: |                                    |
| Deleted: 4A         Deleted: fiber         Deleted: fiber         Deleted: 4B         Deleted: 4C         Deleted: 5         Deleted: 5         Deleted: 4E         Deleted: (P):         Deleted: ,         Deleted: 6B         Deleted: 6E                                                                                   | Deleted: 4A         Deleted: fiber         Deleted: fiber         Deleted: 4B         Deleted: 4C         Deleted: 5         Deleted: 5         Deleted: 4E         Deleted: (P):         Deleted: ,         Deleted: 6B         Deleted: 6E                                                                        | Deleted: | the August 16-17, 2013 deployment, |
| Deleted: fiber
| Deleted: fiber
Deleted: 6B                                                                                                                                                                      | Deleted: fiber           Deleted: 4B           Deleted: 4C           Deleted: 5           Deleted: 5           Deleted: 4E           Deleted: (P):           Deleted: ,           Deleted: 6B           Deleted: 6E                                                                                                 | Deleted: | fiber                              |
| Deleted: 4B
Deleted: 6E                                                                                                                                                           | Deleted: | : fiber                            |
| Deleted: 4C
Deleted: 6B                                                                                                                                                                                                       | Deleted: 4C
Deleted: 6E                                                                                                                                                                                   | Deleted: | : 4B                               |
| Deleted: 5
Deleted: 6E                                                                                                                                                                                                       | Deleted: 5
Deleted: 6E                                                                                                                                                                                            | Deleted: | - 4C                               |
| Deleted: 5
Deleted: 6E                                                                                                                                                                                                                     | Deleted: 5
Deleted: 6E                                                                                                                                                                                                          | Deleted: | : 5                                |
| Deleted: 4E
Deleted: 6E                                                                                                                                                                                                                                   | Deleted: 4E
Deleted: 6E                                                                                                                                                                                                                        | Deleted: | : 5                                |
| Deleted: 4E
Deleted: 6B                                                                                                                                                                                                                                                  | Deleted: 4E
Deleted: 6E                                                                                                                                                                                                                        |          |                                    |
| Deleted: (P):
Deleted: 6E                                                                                                                                                                                                                                                  | Deleted: (P):
Deleted: 6E                                                                                                                                                                                                                                       | Deleted: | : 4E                               |
| Deleted: (P):
Deleted: 6E                                                                                                                                                                                                                                                  | Deleted: (P):
Deleted: 6E                                                                                                                                                                                                                                       |          |                                    |
| Deleted: ,
Deleted: 6E                                                                                                                                                                                                                                                                   | Deleted: ,
Deleted: 6E                                                                                                                                                                                                                                                        | Deleted: | : (P):                             |
| Deleted: (Fig. 6
Deleted: 6E                                                                                                                                                                                                                                                                                 | Deleted: (Fig. 6
Deleted: 6E                                                                                                                                                                                                                                                                      | Deleted: | ,
,                             |
| Deleted: (Fig. 6
Deleted: 6E                                                                                                                                                                                                                                                                                 | Deleted: (Fig. 6
Deleted: 6E                                                                                                                                                                                                                                                                      |          |                                    |
| Deleted: (Fig. 6
Deleted: 6E                                                                                                                                                                                                                                                                                 | Deleted: (Fig. 6
Deleted: 6E                                                                                                                                                                                                                                                                      |          |                                    |
| Deleted: 6B
Deleted: 6E                                                                                                                                                                                                                                                                                                     | Deleted: 6B
Deleted: 6E                                                                                                                                                                                                                                                                                          | Deleted  | · (Fig. 6                          |
| Deleted: 6B
Deleted: 6E                                                                                                                                                                                                                                                                                                     | Deleted: 6B                                                                                                                                                                                                                                                                                                         | Deleteu. | (Fig. 0                            |
| Deleted: 6E                                                                                                                                                                                                                                                                                                                    | Deleted: 6E                                                                                                                                                                                                                                                                                                         | Deleted: | 6B                                 |
| Deleted: 6E                                                                                                                                                                                                                                                                                                                    | Deleted: 6E                                                                                                                                                                                                                                                                                                         |          |                                    |
|                                                                                                                                                                                                                                                                                                                                |                                                                                                                                                                                                                                                                                                                     | Deleted: | 6E                                 |
|                                                                                                                                                                                                                                                                                                                                |                                                                                                                                                                                                                                                                                                                     |          |                                    |
|                                                                                                                                                                                                                                                                                                                                |                                                                                                                                                                                                                                                                                                                     |          |                                    |
|                                                                                                                                                                                                                                                                                                                                |                                                                                                                                                                                                                                                                                                                     | Deleted  | 6F                                 |

artefact was supported by the comparison with sensor Q, which produced the typical circadian cumulative flux pattern with a

- 405 steady increase during the light phase until sunset and decrease thereafter throughout the dark phase. After 5:00, still during dark conditions, the increase in cumulative P flux and divergence from the cumulative Q flux suggested that sensor P then lost its calibration, which occurs when the sensor loses some of the dye coating that produces the signal (e.g. through particle impact). The temporary good agreement of the cumulative fluxes based on P and Q optode readings permitted salvaging sections of the flux record and thereby allowed at least rough estimates for day and nighttime fluxes for this deployment.
- 410 (daytime: 4.3 ± 2.6 mmol m-2h-1, nighttime: -3.2 ± 0.6 mmol m-2h-1, Fig. 7D). The parallel chamber deployments supported these estimates,

**3.2 Differences between P and Q fluxes and comparison with advection chamber fluxes**

After exclusion of flux intervals compromised by biofouling (Case A: no exclusion, Case B: 14:00-18:00, Case C: 12:00 22:00, 5:00-9:00), the differences between P and Q optode fluxes derived from the slopes of the cumulative flux curves (Figs. 3E, 5E, 7E) averaged 2.3%, -0.1% and -4.7% during daytime and 1.7%, 16.2% and -3.2% during nighttime, for the three deployments respectively. These smaller than 20% differences between P and Q optode fluxes strengthened the flux estimates. Fluxes determined with the 2OEC further were supported by the fluxes measured with the advection chambers conducted parallel to the eddy covariance measurement during the August 2013 deployments (Cases A and C, Fig. 8). The average

420 chamber daytime fluxes for the two deployments  $(3.9 \pm 3.0 \text{ mmol m}^2\text{h}^{-1})$  were similar to the respective eddy covariance fluxes  $(3.7 \pm 0.9 \text{ mmol m}^2\text{h}^{-1})$  (Fig 8C), although the chamber nighttime fluxes  $(-3.4 \pm 0.8 \text{ mmol m}^{-2}\text{h}^{-1})$  exceeded those of the eddy covariance instrument  $(-2.5 \pm 1.3 \text{ mmol m}^{-2}\text{h}^{-1})$  by factor 1.4 (Fig 8C). This discrepancy was caused by the smaller nighttime fluxes recorded by the 2OEC during the second August 2013 deployment (Case A, Fig 8B), however, The differences between average eddy and average chamber fluxes were statistically not significant (Fig 8C).

**425 4 Discussion**

430

The small and rapid changes in concentration and flow the aquatic eddy covariance instrumentation must record for accurate flux determination make the technique sensitive to even small disturbances affecting the measuring process (Reimers et al., 2012), By using two oxygen, sensors recording in parallel, the 2OEC allows detection of measuring artefacts and thereby can enhance the reliability of the flux determinations. The functionality of the 2OEC and the ranges of fluxes it recorded were supported by the general agreement between the 2OEC fluxes and advection chamber fluxes measured parallel to the eddy flux recordings.

A perfect conformity of eddy and chamber fluxes cannot be expected due to fundamental differences between the open, noninvasive eddy covariance and the closed, invasive chamber measuring principles. Although marine sandy sediments as those

| l | Deleted: The parallel chamber deployments supported these estimates |
|---|----------------------------------------------------------------------------|
| ( | Deleted: 7D).                                                              |

| Del | eted | :7 |
|-----|------|----|
|     |      |    |

**Deleted: 7C**

**Deleted: 7C**

**Deleted: difference**

Deleted: 7B). The chambers measured flux over the same enclosed sediment area over the duration of the deployment, while the footprint of the eddy measurements moved with current direction, which may explain the differences in dark fluxes observed between 20EC and chambers during the 16-17 Aug 2013 deployment (Fig. 7 B)....

| ( | Deleted: (Reimers et al., 2012b). |
|---|-----------------------------------|
| ( | Deleted: solute                   |
| ( | Deleted: general                  |

investigated here may appear homogeneous, bottom current patterns and patchy colonization by e.g. algae and macrofauna
cause some spatial and temporal variability in organic matter content and associated microbial metabolic activities that may
influence interfacial solute fluxes (Kourelea et al., 2004;Ricart et al., 2015;Wilde and Plante, 2002;Attard et al., 2019). The
fluxes recorded by the 2OEC originated from a sediment surface area of approximately 40 m2 upstream the instrument (Berg
et al., 2007), and the location of this footprint area moved with flow direction. The 2OEC therefore integrated some of the
natural spatial variability of the flux, and the movement of the footprint area as well as changes in bottom flow velocity are
reflected in the measured fluxes. In contrast, each chamber enclosed the same surface area of about 0.03 m2 and applied the
same constant pressure gradient at the sediment surface throughout each deployment. The exclusion of flow variations,
temporal water composition changes, and some of the light affected the fluxes measured by the chambers. These differences
between the two flux-measuring techniques may explain some of the discrepancies in dark fluxes observed between 2OEC
and chambers during the 16-17 Aug 2013 deployment (Fig. 8 B) and emphasize the need for including natural bottom currents

The 2OEC improves detection of sensor fouling. This is significant as the most common and most unnoticed cause for aquatic eddy covariance measuring errors likely is the attachment of marine snow particles or biofilms to the solute sensor (extreme case shown in Fig. 1b). Through physical separation of the sensing surface from the water, such fouling increases sensor response time, which decreases the measured rates of oxygen change and the temporal alignment of oxygen and flow data.

- 470 Furthermore, biological and chemical reactions in such organic coatings can produce or consume oxygen and thereby compromise flux calculations. As the growth of a biofilm on the sensor may be gradual, the detection of the onset of flux bias caused by biofouling may be impossible in a single-sensor instrument. A very good agreement of the cumulative fluxes calculated from the two 2OEC optodes as seen in Case A4 is a strong indication that the sensors measured correctly (Fig. 3), while differences between the cumulative fluxes as observed in Case C4 are indicative of sensor malfunction (Fig. 7E).
- 475 The comparison of the cumulative fluxes can reveal even short or small deviations of the sensor signal as e.g. caused by a temporary attachment of a marine snow particle (Fig. 5E). If one of the two parallel measuring sensors showed a temporary increase or drop in oxygen as found in Cases B and C, we attributed this to the biofouling of that sensor, and in-situ inspections of the sensors confirmed biofouling. Marine snow was present in the water at the study site partly due to its proximity to coral reefs that release mucus to the water (Wild et al., 2004). In Case B, unusual contributions to optode P fluxes in the wave
- frequency band (0.2 to 0.3 Hz) that were not mirrored in optode Q, identified optode P as compromised starting at 15:00 for a ~3 h duration (Fig. Q). A marine snow particle with photosynthesizing organisms attached to the tip of the oxygen sensor P may have caused the erroneous flux estimates. Oxygen concentration in the centre of such aggregates during light conditions can be increased by 85 % relative to the surrounding water (Ploug and Jorgensen, 1999), or even by 180% within the sticky millimetre-size gelatinous colonies of *Phaeocystis spp.*, a common global bloom-forming phytoplankton organism (Ploug et al., 1999). The movement of such an attached photosynthesizing particle, by wave orbital motion can synchronize vertical

| -( | Deleted: in the 16-17 August 2013 deployment |
|----|----------------------------------------------|
| ~( | Deleted: worked                              |
| (  | Deleted: 2                                   |
| (  | Deleted: in the 14-15 August 2013 deployment |
| Y( | Deleted: 6E                                  |
| (  | Deleted: 4E). During                         |

current flow oscillations and the effect of the particle on the oxygen reading (e.g. increased oxygen due to photosynthesis) and thereby lead to erroneous flux estimates (Fig. 9).

In single sensor eddy covariance instruments, obvious temporary sensor malfunctions typically flag long sections or the entire deployment as compromised because it is difficult to determine with certainty when and for how long the sensor reading has

500 been biased. The relatively frequent occurrence of sensor fouling therefore causes substantial losses in data, time and costs. The dual sensor approach can reduce such losses because it allows identifying periods of unbiased measurements within partly compromised data records.

The reliability of the flux data hinges on unbiased sensor data that can capture temporal variability of current flow and the oxygen it carries, which may change as rapidly as 1-3 Hz (McGinnis et al., 2008;Kuwae et al., 2006), The ADV used in the 20EC can produce calibrated current data non-invasively at a frequency of 64 Hz, while the fibre optode has a slower response time (200-300 ms, (Merikhi et al., 2018)), and its placement near the ADV measuring volume could affect current flow measurements and thereby bias the flux calculations. A cylindrical sensor placed in the path of the flow upstream the ADV measuring volume can shed a vortex street thereby compromising the flow in the measuring volume and the flux estimates based on the flow measurements. Depending on the flow Reynolds number, such vortices may extend between 5 to 20 times the diameter of the cylinder downstream the sensor (Green, 2012). By using The Pyroscience fibre optode for the 20EC, one

- of the smallest and fastest oxygen sensors presently available, potential errors caused by the disturbance of the flow and interference with the acoustic pulses of the Doppler velocimeter can be avoided. At the turbulent Reynolds numbers typical for our study site (4000 < Re < 110000), the vortices shed by the 430 µm fibre exposed to the water currents extend between 2 to 10 mm downstream of the fibre. Since the tips were placed at 30 mm horizontal distance from the lower edge of the ADV
- 515 measuring volume, turbulence caused by fibre-flow interaction could not reach the ADV measuring volume. Similarly, the sensor tips at that distance did not interfere with the acoustic pulses of the ADV, and when initially positioning the optode tips, we confirmed that the fibres did not cause any disturbances in the ADV signal.

The Pyroscience fibre optode used with the 2OEC is one of the smallest and fastest oxygen sensors available, and a comparison with the most common oxygen sensors presently utilized for aquatic eddy covariance (Table A4) favours the selection of this sensor for many field settings. For this comparison, three eddy covariance instruments equipped with either (1) one Unisense electrochemical microelectrode (Berg et al., 2019), (2) one JFE Advantech Rinko planar optode (Berg et al., 2016), or (3) one Pyroscience fibre optode were deployed side by side (i.e. 10 m spacing) at our study site 3-4 December 2016. All instruments used the same type of tripod and ADV and the oxygen sensors were mounted at a 45-degree downward angle as described for

525 the 2OEC. The three different sensors measured very similar fluxes when the current flow approached the sensor tips from the front as shown in Fig. 10a, burst 11 to 28. This changed when the flow approached the sensors from the back. The RINKO sensor under such flow conditions may self-shade its planar optode, which may result in an underestimation of the fluxes at

| Deleted: (Kuwae et al., 2006; McGinnis et al., 2008a). |   |
|--------------------------------------------------------|---|
|                                                        | _ |
| Deleted: fiber                                         |   |
|                                                        |   |
| Deleted: may                                           |   |
|                                                        |   |
| Deleted: could                                         |   |

| -( | Deleted: | favors              |
|----|----------|---------------------|
|    |          |                     |
|    |          |                     |
| -( | Deleted: | (Berg et al., 2016) |
| (  | Deleted: | fiber               |

higher frequencies (0.1-1.0 Hz, Fig. 10h) as seen in burst 1 to 9 (Fig. 10a), and possibly also disturb the flow in the Vector flow measuring volume. In environments with unidirectional current, however, the Rinko sensor facing the flow can produce very clean flux data due to its relatively large sensing surface (Berg and Pace, 2017). There were no significant differences between the fluxes based on the fibre optode and the microelectrode for the reversed flow, supporting the choice of the sturdier fibre optodes for oxygen measurements with aquatic eddy covariance instruments in settings with changing flow direction.

**5** Conclusions**

- We propose using the agreement/disagreement between the fluxes calculated from the signals of two independently measuring optodes as a tool to assess the quality of the measured fluxes. The nearly identical cumulative fluxes calculated from the two optodes in our August (Case A) and April (Case B) deployments strongly imply that the dynamics of the fluxes were measured accurately by the system. Likewise, the near linearity of the cumulative flux increase during daytime and decrease during nighttime (Figs. 3e, 5e) and the very similar slopes of these cumulative flux curves support that the measurements recorded representative fluxes. The good agreement of the fluxes measured with the eddy covariance instrument and the fluxes measured
- 550 independently with a very different method (advection chambers, Fig. 8a, b, d) indicate that the magnitudes of the fluxes recorded by the 2OEC were correct. The deployments of the 2OEC in the Florida Keys sandflat revealed that biofouling frequently affects the aquatic eddy covariance measurements even in such an oligotrophic environment with very clear water containing low amounts of phytoplankton, bacteria and particles. Further developments of the aquatic eddy covariance technique therefore may benefit from installations of devices that monitor (e.g. with a camera) and reduce or prevent biofouling
- 555 (e.g. through a cleaning mechanism). This project intended improving the reliability of the aquatic eddy covariance technique and the procedures of data analysis in order to promote this powerful technique. The advantages of 20EC flux measurements over invasive measurements (e.g. benthic chambers) may be most significant for deployments in continental margins. The magnitudes of biogeochemical benthic processes increase with decreasing water depth, with benthic fluxes reaching highest rates and dynamics in the shelf environment (Huettel et al., 2014;Middelburg and Soetaert, 2004;Jahnke, 2010;Bauer et al.,
- 560 2013;Reimers et al., 2004). Here lighta bottom currents and waves may strongly influence benthic fluxes (Gattuso et al., 2006). The relatively high fluxes and daytimea oxygen release recorded at our oligotrophic sandy study site, supported by flux measurements from similar subtropical and tropical carbonate environments (Bednarz et al., 2015;Rao et al., 2012;Wild et al., 2009;Wild et al., 2005;Glud et al., 2008), emphasize the need for instrumentation that reliably can take light and flowa at the seafloor into account when measuring benthic fluxes. The 20EC is a powerful tool that meets these requirements, and its
- 565 relatively high temporal resolution can provide new insights into the dynamics of benthic oxygen flux.

| Deleted: | 8B    |
|----------|-------|
| Deleted: | 8A    |
|          |       |
|          |       |
| Deleted: | fiber |
| Deleted: | fiber |

| Deleted: | the     |
|----------|---------|
| Deleted: | and     |
| Deleted: | benthic |

580

1

**6 Appendix A**

585 Table A1: Specifications of the Pyroscience™ FireStingO2-Mini oxygen meter . This oxygen meter recently has been replaced by the Pyroscience™ PICO-O2, which is similar to the FireStingO2-Mini but more compact. The FireStingO2-Mini oxygen meter is still available in combination with the Pyroscience™ subport (FSO2-SUBPORT).

| Pyroscience ™ FireStingO 2 -Mini | Single sensor module,                                                |                |
|--------------------------------------------------------|----------------------------------------------------------------------|----------------|
| Oxygen port                                            | 1 fibre-optic ST-connector                                           | Deleted: fiber |
| Temperature port                                       | 4-wire PT100, -30°C-150°C, 0.02°C resolution, ±0.5°C accuracy        |                |
| Dimensions and Weight                                  | 67 x 25 x 25 mm, 70 g                                                |                |
| Measuring principle                                    | Luminescence lifetime detection (REDFLASH)                           |                |
| Excitation Wavelength                                  | 620 nm (orange-red)                                                  |                |
| Emission wavelength                                    | 760 nm (NIR)                                                         |                |
| Maximum sampling rate                                  | 20 Hz                                                                |                |
| Interface                                              | Serial interface (UART), ASCII communication protocol                |                |
| Analog output                                          | 0 - 2.5 V DC, 14 bit resolution (the meter presently uses half the   |                |
|                                                        | potential voltage range of the Vector analogue input, but            |                |
|                                                        | developments are in place to increase this to the full range (0-5V). |                |
| Power requirements                                     | Max. 70 mA at 5 V DC from USB (typ. 50 mA)                           |                |

**Table A2: Specifications of the Pyroscience™ OXR430-UHS retractable oxygen minisensors**

| Table 12. Specifications of the Tyroscience 0 | Artes of the retractable oxygen miniscusors |   |   |                |
|-----------------------------------------------|---------------------------------------------|---|---|----------------|
| Optical O 2 fibre, sensor type     | Pyroscience ™ OXR430-UHS         |   |   | Deleted: fiber |
| Fibre, diameter                               | 430 μm                                      |   | ( | Deleted: Fiber |
| Optimal measuring range                       | 0-720 μmol l -1                  |   | × |                |
| Maximum measuring range                       | 0 - 1440 μmol l -1               |   |   |                |
| Response time                                 | < 0.3 s                                     |   |   |                |
| Detection limit                               | 0.3 µmol 1-1                                |   |   |                |
| Resolution at 1% O 2               | 0.16 µmol 1 -1                   |   |   |                |
| Resolution at 20% O2                          | 0.78 μmol 1 -1                   |   |   |                |
| Accuracy at 1% O 2                 | $\pm 0.31 \ \mu mol \ l^{-1}$               |   |   |                |
| Accuracy at 20% O 2                | $\pm 3.13 \ \mu mol \ l^{-1}$               |   |   |                |
| Temperature range                             | 0 - 50°C                                    | ] |   |                |

590

**Table A3: Specifications of the NORTEK Vector acoustic Doppler velocimeter**

| Sensor      | Range                                         | Accuracy          | Precision/Resolution   |
|-------------|-----------------------------------------------|-------------------|------------------------|
| Velocity    | ±0.01, 0.1, 0.3, 1, 2, 4, 7 m s -1 | $\pm 0.5\%$       | $\pm 1\%$              |
| Pressure    | 0-20 m (shallow water version)                | 0.5% (full scale) | < 0.005% of full scale |
| Temperature | -4 to +40 °C                                  | 0.1 °C            | 0.01 °C                |
| Compass     | 360°                                          | 2°                | 0.1°                   |
| Tilt        | < 30°                                         | 0.2°              | 0.1°                   |
|             |                                               |                   |                        |

| 1   | Table A4. The specifications of                                                                                                                                                 | the oxygen microelectrode, Rink                                                                                                     | o planar optode and Pyroscie                                                                                                               | ence fibre optode                                                                                                                                                                 | Deleted: fiber             |
|-----|---------------------------------------------------------------------------------------------------------------------------------------------------------------------------------|-------------------------------------------------------------------------------------------------------------------------------------|--------------------------------------------------------------------------------------------------------------------------------------------|-----------------------------------------------------------------------------------------------------------------------------------------------------------------------------------|----------------------------|
| -   | Sensor                                                                                                                                                                          | OX-10 fast (µm)                                                                                                                     | RINKO EC                                                                                                                                   | OXR430-UHS                                                                                                                                                                        |                            |
|     | Туре                                                                                                                                                                            | Microelectrode                                                                                                                      | Planar optode                                                                                                                              | Fibre optode                                                                                                                                                                      | Deleted: Fiber             |
|     | Manufacturer                                                                                                                                                                    | Unisense                                                                                                                            | JFE-Advantech                                                                                                                              | Pyroscience                                                                                                                                                                       |                            |
|     | Measurement principle                                                                                                                                                           | Electrolytical reduction                                                                                                            | Phosphorescence                                                                                                                            | Phosphorescence                                                                                                                                                                   |                            |
|     | Tip diameter (µm)                                                                                                                                                               | 10                                                                                                                                  | 12000                                                                                                                                      | 430                                                                                                                                                                               |                            |
|     | Response time (90%) (s)                                                                                                                                                         | < 0.3                                                                                                                               | < 0.5                                                                                                                                      | < 0.3                                                                                                                                                                             |                            |
|     | Range (% air saturation)                                                                                                                                                        | 0-200                                                                                                                               | 0-200                                                                                                                                      | 0-500                                                                                                                                                                             |                            |
| 600 | 7 Data availability
The current flow and oxygen of
at the Biological and Chemica
10.26008/1912/bco-dmo.8125
concentrations recorded by the
Chemical Oceanography | data collected with the 2OEC du
al Oceanography Data Managem
523.1. Suggested Citation: Huet
he 2OEC-instrument in the Flo | tring the August 2013 and A
tent Office (BCO-DMO, htt
tel, M., Berg, P., Merkihi,
prida Keys from August 20
(BCO DMO) (Varging | April 2014 deployments are available
ps://www.bco-dmo.org/) under DOI:
A. (2020) Current flow and oxygen
113 and April 2014. Biological and
1) Varsion Data 2020.05.2 | Deleted: were submitted to |
|     | Cnemical Oceanography Data Management Office (BCO-DMO). (Version 1) Version Date 2020-05-2.                                                                                     |                                                                                                                                     |                                                                                                                                            |                                                                                                                                                                                   |                            |
| 605 | DOI:10.26008/1912/bco-dmo                                                                                                                                                       | .812523.1 [access date]                                                                                                             |                                                                                                                                            |                                                                                                                                                                                   | Deleted: ).                |
|     | 8 Author contributions
MH designed and assembled t
the manuscript with contribution                                                                                       | he 20EC instrument, MH, PB a
ions from all co-authors.                                                                           | nd AM deployed the 20EC,                                                                                                                   | and analysed the data. MH prepared                                                                                                                                         | Deleted: analyzed          |
|     | 9 Competing interest statem                                                                                                                                                     | ent                                                                                                                                 |                                                                                                                                            |                                                                                                                                                                                   |                            |
| 610 | ) The authors declare that they have no conflict of interest.                                                                                                                   |                                                                                                                                     |                                                                                                                                            |                                                                                                                                                                                   |                            |
|     | 10 Acknowledgments                                                                                                                                                              |                                                                                                                                     |                                                                                                                                            |                                                                                                                                                                                   |                            |
|     | We thank the staff of the FIO                                                                                                                                                   | Florida Keys Marine Laborato                                                                                                        | ry for help with instrument                                                                                                                | deployments and sample collection.                                                                                                                                                |                            |
| 1   | We also would also like to the                                                                                                                                                  | ink DSO Chris Peters and the st                                                                                                     | off of the FSU coastal marin                                                                                                               | e lab for providing SCUBA support                                                                                                                                                 |                            |
|     | Brian W. Wells, Pascal Brigr                                                                                                                                                    | nole, Lee Russell, and Natalie (                                                                                                    | Geyer helped with the cham                                                                                                                 | ber deployments. We would like to                                                                                                                                                 |                            |
| 615 | extend our special thanks to re-                                                                                                                                                | eferees Clare Reimers and Karl                                                                                                      | Attard. Their questions and                                                                                                                | comments greatly helped to improve                                                                                                                                                |                            |
|     | the first version of this man                                                                                                                                                   | uscript. The research was con                                                                                                       | ducted under NOAA per                                                                                                                      | nit FKNMS-2012-137-A2 and was                                                                                                                                                     |                            |
|     |                                                                                                                                                                                 |                                                                                                                                     |                                                                                                                                            |                                                                                                                                                                                   | Deleted: and               |
|     | supported by NSF grants OCF                                                                                                                                                     | E-1334117, OCE-1851290, and                                                                                                         | UCE-1061364.                                                                                                                               |                                                                                                                                                                                   | Deleted                    |
|     |                                                                                                                                                                                 |                                                                                                                                     |                                                                                                                                            |                                                                                                                                                                                   | Deletea:                   |
|     |                                                                                                                                                                                 |                                                                                                                                     | 14                                                                                                                                         |                                                                                                                                                                                   |                            |

[revised manuscript text omitted]

  - 20